# An improved SNAP-ADAR tool enables efficient RNA base editing to interfere with post-translational protein modification

Karthika Devi Kiran Kumar[1], Shubhangi Singh [1], Stella Maria Schmelzle [1], Paul Vogel[2], Carolin Fruhner[1], Alfred Hanswillemenke [1], Adrian Brun[1], Jacqueline Wettengel[1], Yvonne Füll [1], Lukas Funk [1], Valentin Mast[1], J. Josephine Botsch [1], Philipp Reautschnig [1], Jin Billy Li [2] & Thorsten Stafforst [1,3,4] ✉

RNA base editing relies on the introduction of adenosine-to-inosine changes into target RNAs in a highly programmable manner in order to repair disease-causing mutations. Here, we propose that RNA base editing could be broadly applied to perturb protein function by removal of regulatory phosphorylation and acetylation sites. We demonstrate the feasibility on more than 70 sites in various signaling proteins and identify key determinants for high editing efficiency and potent down-stream effects. For the JAK/STAT pathway, we demonstrate both, negative and positive regulation. To achieve high editing efficiency over a broad codon scope, we applied an improved version of the SNAP-ADAR tool. The transient nature of RNA base editing enables the comparably fast (hours to days), dose-dependent (thus partial) and reversible manipulation of regulatory sites, which is a key advantage over DNA (base) editing approaches. In summary, PTM interference might become a valuable field of application of RNA base editing.

RNA base editing refers to the reprogramming of genetic information at the RNA level. In mammals, accessible changes include the post-transcriptional deamination of adenosine to inosine (A-to-I editing) and cytidine-yielding uridine (C-to-U editing), which are carried out by the enzymes from the ADAR (adenosine deaminase acting on RNA) and the APOBEC (apolipoprotein B mRNA editing enzyme, catalytic polypeptide) families, respectively[1–3]. As inosine is biochemically interpreted as guanosine, A-to-I editing results formally in A-to-G changes in the genetic information[4,5]. Since 2012, ADAR activity has been engineered into numerous tools for targeted RNA base editing[6–10]. Most tools employ guide RNAs, which are linked to engineered ADAR fusion proteins, and recruit them to target sites in RNAs[6,7,9]. Given ADAR's rigorous specificity for double-strand RNA substrates, the guide RNA allows to define the targeted adenosine by forming the guide RNA/target RNA duplex in a rationally programmable manner with very high precision. The numerous tools vary particularly in the way the ADAR fusion protein is connected to the guide RNA. In the SNAP-ADAR tool, the deaminase domain is fused to a SNAP-tag, which enables the covalent attachment to the ca. 20 nt long guide RNA that is required to be chemically modified, including a self-labeling moiety, e.g., O6-benzylguanin (BG), to assist the self-labeling reaction[11,12]. This covalent in-situ assembly reaction is very fast and efficient, and can be run in vitro, in vivo and can be put under control of light[13–15]. Since the guide RNAs of the SNAP-ADAR approach need to carry the BG moiety, guide RNAs cannot be genetically encoded but need to be transfected into cells and are typically chemically stabilized by terminal phosphorothioate linkages and global 2′-O-methylation keeping only a window of three nucleotides unmodified[11,12]. This central base triplet

¹Interfaculty Institute of Biochemistry, University of Tübingen, Tübingen, Germany. ²Department of Genetics, Stanford University, Stanford, CA, USA. ³Gene and RNA Therapy Center (GRTC), Faculty of Medicine University Tübingen, Tübingen, Germany. ⁴iFIT Cluster of Excellence (EXC2180) "Image-Guided and Functionally Instructed Tumor Therapies, University of Tübingen, Tübingen, Germany. ✉e-mail: thorsten.stafforst@uni-tuebingen.de

defines a small window for editing, thus enabling a very efficient control of bystander off-target edits on the target RNA[12]. However, all engineered editing tool have in common that the ectopic expression of ADAR fusion proteins elicits notable, largely guide RNA-independent global off-target effects[16,17]. This is particularly true for fusion constructs that harbor a hyperactive E > Q ADAR mutation[18]. Most RNA base editing tools currently apply the E > Q mutation, as it allows to improve editing efficiency for less favored substrates, e.g., substrates which carry a guanosine or cytidine nucleotide 5′ to the target adenosine (5′-SAN, S = C, G; $\underline{A}$ = target adenosine, N = any nucleotide). Furthermore, in the context of the SNAP-ADAR tool, the Q-mutant strongly improves the potency of the guide RNA (by ca. 10-fold)[12], and it can be expected that the latter is true also for other RNA base editing tools. To simplify the tool, but also to strongly reduce global off-target editing, the SNAP-ADAR tool is typically integrated into the genome of the host cell and editing is started and controlled by transfection of one or several guide RNAs[12]. On endogenous signaling transcripts like *KRAS* or *STAT1*, editing yields >50% were achieved in cell culture when the hyperactive SNAP-ADAR1 E > Q (SA1Q) tool was used, and sufficient guide RNA amounts (20 pmol/96-well scale) were transfected[12].

While most RNA base editing tools are currently being explored for the reversal of disease-causing G-to-A point mutation, it was already early postulated that RNA base editing could be used to interfere with signal transduction and metabolism[6,19,20]. On one hand, this is driven by the scope of amino acid substitutions which are accessible with A-to-I base editing. Specifically, almost all amino acids which represent commonly used regulatory sites of posttranslational modification (PTM) can be defunctionalized by RNA base editing. These include the reversibly phosphorylated amino acids tyrosine, threonine, and serine, which can be reprogrammed to cysteine, alanine, and glycine, respectively, and the basic amino acid lysine, which can be reprogrammed to arginine, to suppress lysine methylation, acetylation or ubiquitination/sumoylation[20]. On the other hand, the doseable and reversible nature of RNA base editing is ideally suited to transiently interfere with signaling cues and metabolism, with respect to essential biochemical pathways, which show adverse effects with permanent changes[21,22]. Such processes, including tissue repair, immune signaling, essential metabolic homeostasis, or epigenetic regulation, are commonly controlled by reversible posttranslational modifications (PTMs) on the above-mentioned amino acids, which are amenable for RNA base editing.

The JAK/STAT pathway, a commonly studied signaling cascade, involves various key post-translational modification events. It is an essential and versatile pathway that is involved in innate and adaptive immunity, cell division, cell death, and homeostasis[23,24]. In response to interferon (IFN), the JAK/STAT pathway activates a set of antiviral and antimycobacterial genes called interferon-stimulated genes (ISGs)[25–27]. More specifically, the recognition of IFN-α results in downstream phosphorylation of STAT1 and STAT2 at specific tyrosine residues resulting in formation of the heterotrimeric ISGF3 (Interferon-stimulated gene factor 3) complex, which activates ISGs that carry an interferon-stimulated response element (ISRE) in their promotor and induce a strong antiviral response[28]. On the other hand, IFN-γ stimulates the phosphorylation and homodimerization of STAT1, the so-called GAF complex (gamma-activated factor), which localizes to the nucleus to activate ISGs that carry a GAS promotor sequence (gamma-activated sequence) and that support defense against mycobacterial infections[29]. In addition to promotor binding, STAT proteins shape gene expression by binding to intronic and intergenic enhancer regions[30]. Loss-of-function (LOF) and gain-of-function (GOF) mutations in STAT1 and other players of the JAK/STAT signaling pathway are well established causes of genetic diseases[31–33]. Dominant LOF mutations of STAT1 that suppress phosphorylation of *Y701* in *STAT1*, strongly reduce IFN-γ response, and lead to mendelian susceptibility to mycobacterial diseases

(MSMD) and osteomyelitis with already one affected allele, while the response to IFN-α and antiviral defense seems comparably unaffected[34]. Dominant GOF mutations are also well characterized for STAT1. Such mutations often affect the phosphorylation state of Y701 in STAT1 indirectly, leading to an increase and prolongation of nuclear and transcriptionally active pSTAT1 (phosphorylated STAT1), and to a dominant positive effect on GAF activation (IFN-γ response), with almost no effect on ISRE activation (IFN-α response)[35]. Counter intuitively, patients carrying dominant GOF mutations in STAT1 suffer from chronic mucocutaneous candidiasis (CMC), which results from an impaired T-cell-mediated IL-17 immunity. Affected individuals typically also show autoimmunity phenotypes depending on the respective GOF mutation[35]. While the antiviral defense was initially described to be normal[34], a recent study on the clinical picture of STAT1 GOF patients revealed an increased susceptibility for viral infections, cancer, and aneurysm[36].

In this work, we introduce an improved version of the SNAP-ADAR tool and apply it successfully to modulate PTM sites in various proteins and show functional interference with JAK/STAT signaling. Overall, PTMi appears as an attractive field of application in which RNA base editing can fully exploit its unique properties.

## Results

### Improved tool performance by guide RNA design

To improve the utility of the SNAP-ADAR approach, we revisited the guide RNA (gRNA) design, starting from the prior design[12] that comprises a guide RNA of 22 nt (19 nt antisense + 3 nt non-binding loop), chemically stabilized by 2′-O-methyl ribose modification outside the central base triplet, and which is equipped with a 5′-terminal, single *O6*-benzylguanine (BG) moiety for covalent SNAP-ADAR recruitment[12] (Fig. 1a). In an initial screen, we identified three key guide RNA properties to improve editing: (1) increasing the length of the guide RNA to ca. 25 nt (22 nt antisense + 3 nt non-binding loop), (2) the inclusion of up to four locked nucleic acid (LNA) building blocks, and (3) the application of a bivalent linker for recruitment of two SNAP-ADAR proteins per guide RNA (BisBG), see Fig. 1a. We have shown the notable effects of these measures in direct comparison to the prior design (BG 22 nt) on the editing of the phospho-tyrosine site 701 on endogenous *STAT1* in engineered 293 HEK cells expressing SNAP-ADAR1Q (Flp-In T-REx 293 – SA1Q), see Fig. 1b. While the prior design required the transfection of 20 pmol/96-well for its maximal editing yield of 80%[12], the same was already achieved with 1 pmol/96-well with the improved design. The highest potency was achieved with the combination of a specific terminal LNA pattern, the BisBG recruiting moiety, and the extension of the guideRNA length to 25 nt, which gave high editing efficiency (65%) with only 0.1 pmol/96-well and detectable editing even at 0.01 pmol/96-well. The Flp-In T-REx 293 – SA1Q cell line allows to fine-tune the SA1Q expression level by the duration of doxycycline induction[12]. In contrast to the prior design, where at least 5 pmol guide RNA (per 96-well) were required to induce moderate editing yield at full SA1Q induction (48 h doxycycline, Fig. 1c), the best design achieved considerably higher editing yields (60-80%) at one tenth of the dose (0.5 pmol/well) and under weaker SA1Q induction (down to 4 h doxycycline). We used Western blot to monitor the amount and formation of guideRNA- mediated covalent SA1Q dimers (Fig. 1d) and found that 0.5 pmol/96-well guide RNA engaged almost all available SA1Q for conjugation upon weak SA1Q induction (8 h doxycycline). Furthermore, the application of the BisBG moiety helped to boost editing of notoriously difficult to edit codons like 5′-G$\underline{A}$N (N = G, A, U, C)[12] in the 22 nt guide RNA design (Fig. 1e). Finally, the improved guide RNA design enabled better usage of the wildtype SNAP-ADAR enzymes, SA1 and SA2, which are more precise regarding global off-target effects, but which have been roughly tenfold less potent in combination with the prior guide RNA design compared to their hyperactive analogs SA1Q and SA2Q[12].

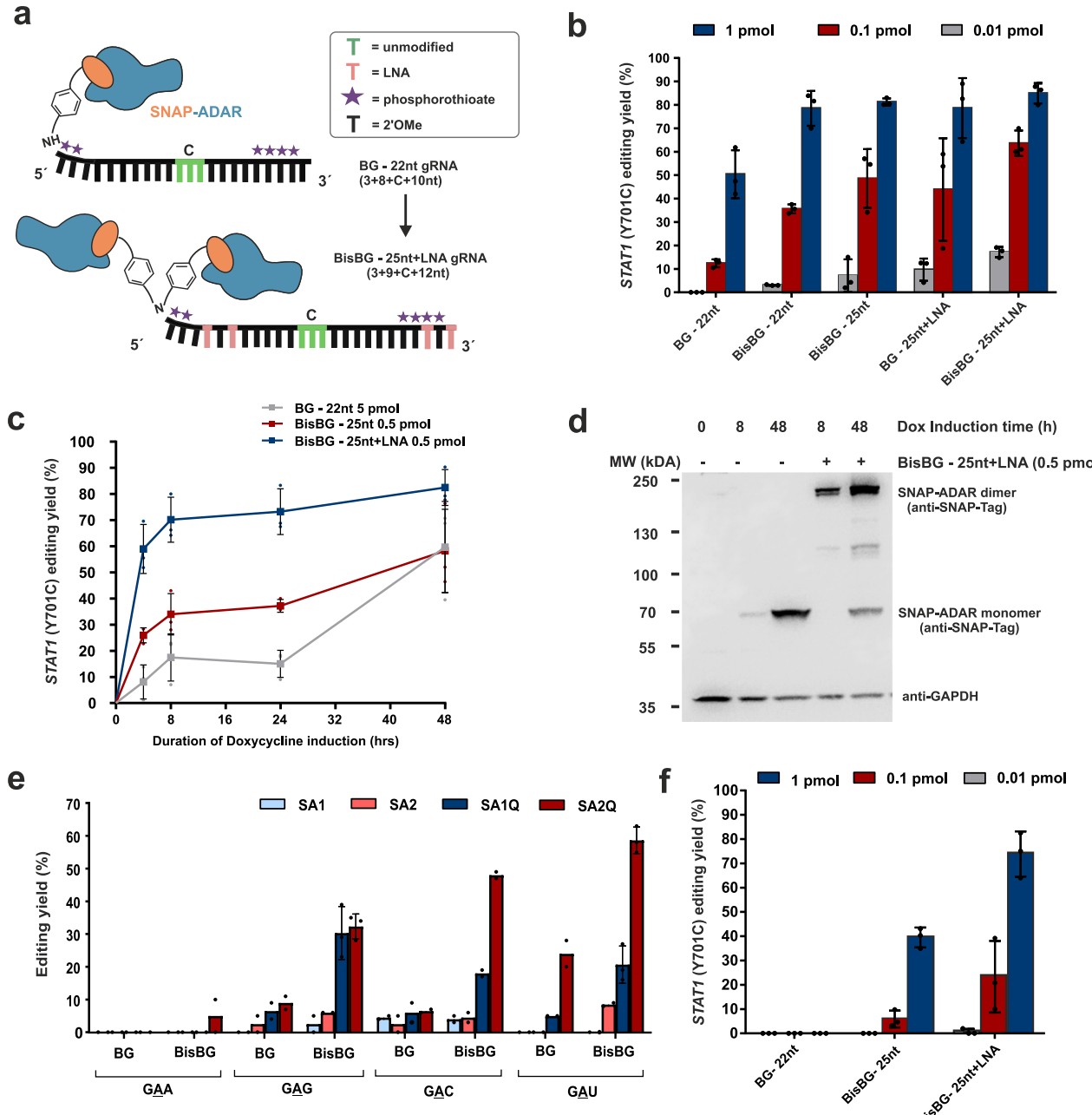

**Fig. 1 | Improved guide RNA design and characterization in engineered 293 Flp-In T-REx cells expressing one of the four SNAP-ADAR effectors (see key). a** The improved guide RNA (gRNA) design is made longer with four LNA s at specific positions (light pink) and can recruit two SNAP-ADAR proteins (BisBG linker in place of mono BG linker as in the prior design). The chemically unmodified nucleotides in the central base triplet are indicated in green, where C denotes the cytidine opposite the targeted adenosine. **b** Editing of 5′-UAU site in the endogenous *STAT1* transcript (*Y701*) comparing different amounts of different guide RNA designs tested (see key) in Flp-In T-REx 293 – SA1Q cells induced with 10 ng/ml of Doxycycline for 48 h. **c** Editing potency of improved guide RNA designs at ten times lower amount compared to old design under varied expression levels of the SA1Q effector by varying doxycycline (Dox) induction times as indicated. **d** Western blot showing unconjugated SNAP-ADAR (SNAP-ADAR monomer) and BisBG-guide RNA-mediated, intracellular covalent dimerization of SA1Q (see band shift, SNAP-ADAR dimer) at low (8 h Dox) and high (48 h Dox) expression levels of the editing enzyme in Flp-In T-Rex 293 – SA1Q cells; Biological replicate blots of high SA1Q expression levels are available in the Supplementary Fig. 1. **e** Editing of 5′-GAN codons in the ORF of endogenous GAPDH transcripts (targets used in Vogel et al.[12]) with 5 pmol of short 22 nt guide RNAs with linkers that could recruit either one (BG) or two (BisBG) SA1Q effectors in Flp-In T-Rex 293 – SA1Q cells (48 h Dox). **f** the same as panel **b**, but using wildtype SA1-expresing Flp-In T-REx 293 cells. Data in panel **b**, **c**, and **f** is shown as the mean ± s.d. of N = 3 independent experiments, and in panel **e**, is shown as the mean ± s.d. of N = 2 (for most cases and N = 3 in some cases) individual data points are shown as dots. All guide RNA amounts denote pmol/96-well with 150 μl total volume. Source data and full western blots are provided as a Source Data file.

Notably, the improved guide RNA design allows to obtain good editing yields (up to 70%) also with the wildtype SA1 tool when applying guide RNA amounts (1 pmol/96-well), that gave no detectable editing with the prior guide RNA design (Fig. 1f), highlighting the benefits of the improved guide RNA design.

## The SNAP-ADAR system is highly versatile in cell culture applications

Due to the high transfection and knockdown efficiencies, chemically synthesized siRNAs still dominate most cell culture applications over genetically encoded shRNAs. Given the small size of the SNAP-ADAR

guide RNAs, their transfection into challenging cells including primary cells might be a strong advantage over genetically encoded editing approaches which may require viral delivery of the guide RNA component. To extend the scope and improve the versatility of the SNAP-ADAR approach, we demonstrate the delivery of the SNAP-ADAR transgene by various means into various cell types and provide proof of successful editing of the endogenous *STAT1* transcript by subsequent guide RNA transfection. Specifically, we applied the PiggyBac transposase system[37] and lentiviral delivery[38,39] for stable integration, and adenoviral delivery[40] for transient expression in HeLa and A549 cells (Fig. 2a), of which the latter cell line is hard to transfect with plasmids. We were able to easily express the transgene in HeLa and A549 cells and to induce maximal editing yields (between 50% and 90%) on the regulatory *STAT1 Y701* phospho site with very good guide RNA potency using the improved guide RNA design (Fig. 2b, c). Since PiggyBac integration is particularly simple and efficient, we studied the potency of the improved guide RNA design more deeply in A549 and HeLa cells stably expressing either wildtype SA1 or hyperactive SA1Q (Fig. 2d, e). In HeLa cells, very high editing yields ≥80% were obtained with SA1Q and guide RNA amounts ≥0.5 pmol (per 96-well). The HeLa cell line expressing the wildtype SA1 effector achieved slightly reduced editing levels (ca. ≥60%) and required slightly more guide RNA (≥1 pmol/96-well) for this. In A549 cells, similar trends were observed, however, with slightly reduced maximal editing yield and guide RNA potency. Interestingly, we found a certain reduction in editing yield for the SA1 effector in A549 cells at high guide RNA amounts. We speculate that the intracellular guide RNA levels at very high doses might use up the entire SNAP-ADAR protein which could reduce the frequency of recruiting two SNAP-ADAR effectors per guide RNA. We used a BG-FITC labeling assay to determine the amount of free, unconjugated SNAP-ADAR protein depending on the guide RNA dose applied (Fig. 2f). Indeed, we found that all effector was used up when ≥5 pmol/ 96-well BisBG-guide RNAs were applied.

Depending on the turnover rate of a given protein, a recoding editing event might take some time to fully establish a phenotype. Thus, we tested the time course of the editing yield with the improved guide RNA design in SA1Q PiggyBac HeLa cells (HeLa-PB-SA1Q). In quickly dividing HeLa cells, the editing yield started to drop after three days (Fig. 2g). To generate a window large enough to establish an editing phenotype at the protein level, we developed a protocol with a second transfection 48 h after the first transfection. This double transfection protocol enables to characterize the edited protein on days 4-7 after the first transfection and should satisfy many applications. If required, a third transfection would also be possible (Fig. 2g).

Finally, we explored the editing of the endogenous *STAT1* transcript in human primary cells by adenoviral delivery of SA1Q (25 to 75 MOI) and transfection of the guide RNA (0.2 – 5 pmol/96-well), see Fig. 2h. In retinal pigment epithelium cells (RPE), moderate editing levels (50-60%) were already achieved at low guide RNA dose (0.2 pmol/96-well). In normal human astrocytes (NHA), the editing levels were generally lower, and higher titer of adenovirus helped to increase the editing yield. Overall, the data highlights the versatility of the SNAP-ADAR tool. Following simple protocols, high editing levels are regularly achieved with low-dose guide RNA transfections. The effector can either be stably expressed in immortalized cell lines by PiggyBac integration, but also transiently and readily in primary cells using adenovirus even without FACS sorting.

## PTM interference: using RNA base editing to perturb protein function

Even though RNA base editing is currently limited to A-to-I (and C-to-U) changes, there are various applications conceivable beyond the repair of disease-causing G-to-A (and T-to-C) point mutations. RNA editing has the potential to modulate native protein function[6,19,20]. This becomes particularly clear when looking into the scope of amino acid changes, which include the removal of (regulatory) phospho-tyrosine (Y > C), -serine (S > G), and -threonine (T > A), as well as the removal of (regulatory) lysine residues (K > R, or K > E). The (reversible) post-translational modification (PTM) of proteins is a hallmark of regulating signaling cues, metabolism, transcription, epigenetics, protein degradation, and many other processes[6,19,20]. RNA base editing could be employed to interfere with posttranslational regulation of protein function in a highly rational and programmable manner. In analogy to RNA interference (RNAi)[41], we call this broader concept PTM interference (PTMi). Applying RNA base editing for PTMi might be an attractive way to study basic biology and to create clinically desirable phenotypes (Fig. 3a). To get an idea about the scope of RNA base editing for PTMi, we set-up an unbiased screen of >70 different PTM sites on various endogenous signaling transcripts (Fig. 3b). For all >70 sites, we transfected guide RNAs of the improved standard design into HeLa-PB-SA1Q cells, which comprised of a 22 nt antisense part (5′−9-C-12 nt, with C = the cytidine opposite the target adenosine), plus a 3 nt 5′-overhang, including 4 LNA bases, and a 5′-terminal BisBG moiety for recruiting two SA1Q effectors per guide RNA. With the improved standard design, we observed editing yields ranging from 0% to 90%, with roughly 40% of the sites being edited with yields ≥50%, highlighting that most of the codons relevant for PTMi are well editable in principle. To better understand the factors that affect editing yields, we analyzed the data further. There is a well-known codon preference for the editing of any 5′-NAN codon (N = A, U, G, C), particularly preferring U > A > C > G for the 5′-neighboring nucleotide. Indeed, we obtained the highest editing yields for 5′-UAG, 5′-UAU and 5′-UAC codons, medium editing levels for 5′-AAN, while 5′-CAN and 5′-GAN (N = A, U, G, C) were clearly more difficult to edit. Interestingly, for each specific codon we found examples with high editing yield but also with rather low editing yield (Fig. 3c). This large spreading in editing efficiency indicates that further key factors play important roles, which could be RNA secondary structure (target or guide RNA), the target gene expression level, or the half-life of the target transcript. To assess structural determinants, we plotted all editing yields against the G/C-content of the guide RNA/target mRNA substrate duplex (Fig. 3d) and against the free energy of guide RNA hairpin folding (Fig. 3e). Indeed, editing yields were higher for guide RNAs with less G/C-content and with little propensity for hairpin folding. We then plotted the editing yields against the expression levels of the target genes (Fig. 3f), using the TPM (Transcript per million) values of the target transcripts determined in the HeLa-PB-SA1Q cell line. We also plotted the editing yields against the average half-lives of the transcripts (Fig. 3g), which have been determined by others in HeLa cells before[42]. Neither the expression level nor the target half-life seems to have a major influence on the editing outcome. Thus, key parameters regarding editing efficiency are the codon preference, the target structure and the guide RNA hairpin folding propensity.

Conveniently, these key parameters can be improved by means of optimizing guide RNA sequence and chemistry. While Tyr>Cys (5′-UAY) and Lys>Arg (5′-AAR) editing gave already satisfying yields, other PTMi targets including Ser>Gly and Thr>Ala editing often suffer from low yields, for example, when 5′-CAN codons were addressed (Fig. 3c). From a recent structural analysis of ADAR2 binding to a dsRNA substrate it was discovered that a clash of the exocyclic amino group of the guanine base with the backbone of glycine residue 489 is responsible for loss of editing efficiency at these un-preferred codons[43]. Thus, we aimed for reducing the space demand of the 5′G:C base pair in the minor grove by applying the nucleoside inosine in the central base triplet for pairing the cytosine base 5′ to the targeted adenosine in 5′-CAN codons, to exemplify: change the 5′-NCG by a 5′-NCI sequence in the guide RNA. We tested the concept on nine endogenous targets, covering all four 5′-CAN codons (N = A, U, G, C). In all nine cases, a deoxyinosine placed at the respective site gave improved editing yields (Fig. 4a). For all four

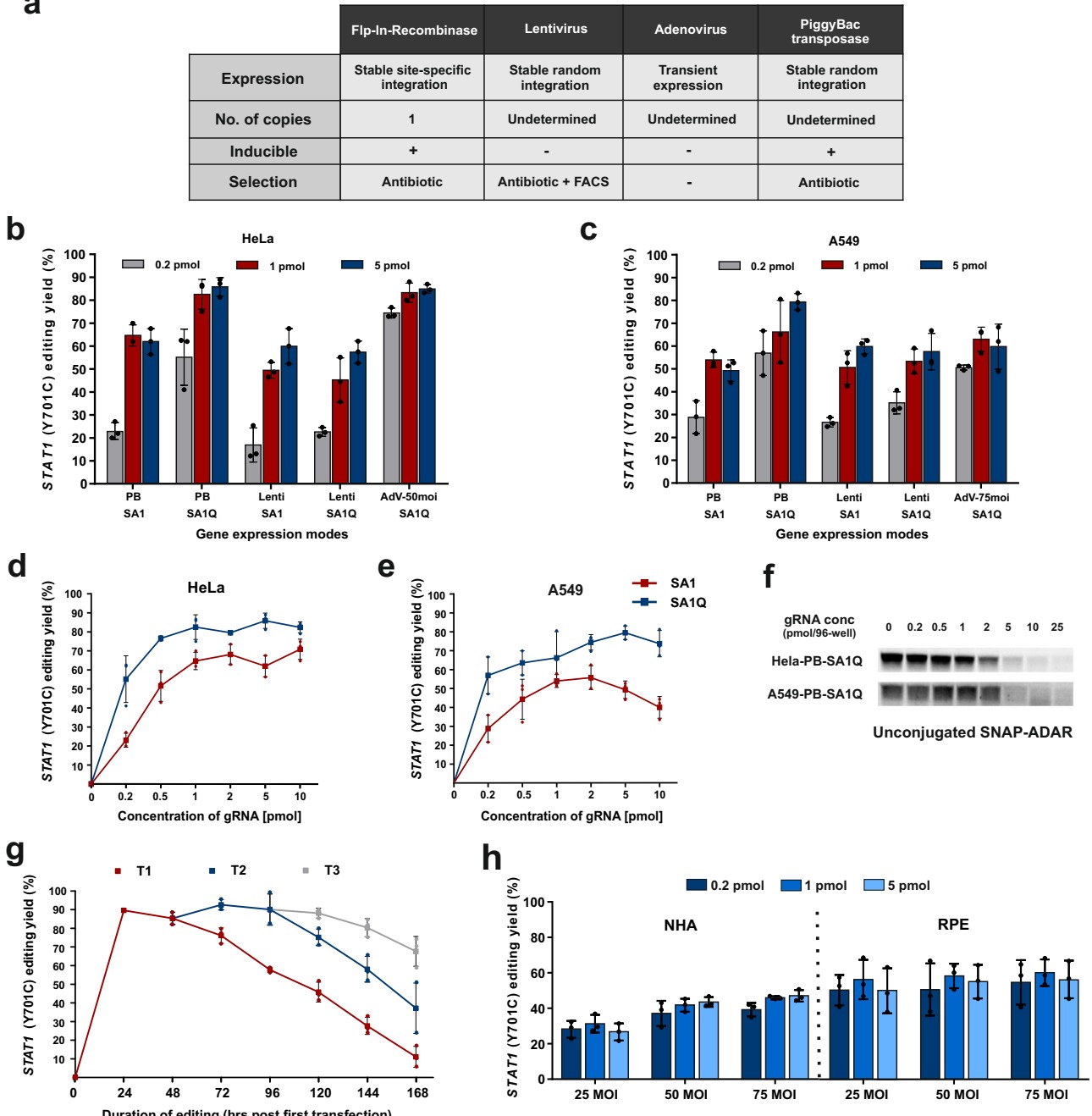

**Fig. 2 | Efficiency of the improved SNAP-ADAR tool in immortalized cell lines and human primary cells. a** Characteristics of the transgene expression systems used to express SNAP-ADAR effector in different cell types. **b**, **c** Comparative editing of 5′-UAU site in the endogenous *STAT1* transcript (*Tyr701*) with different amounts of BisBG – 25nt+LNA guide RNA (BisBG 180) in HeLa (**b**) and A549 (**c**) cells expressing SA1Q or SA1 by different delivery options. **d**, **e** Dose-dependent editing of the *STAT1* transcript (*Y701* site) with BisBG 180 in HeLa-PB-SA1Q/SA1 (**d**) and A549-PB-SA1Q/SA1 (**e**) cells expressing SA1Q or SA1 after full induction (48 h) with 1 µg/ml doxycycline. **f** BG-FITC-protein/gRNA conjugation assay to determine unconjugated SA1Q levels in Hela-PB-SA1Q/A549-PB-SA1Q cells induced with Dox for 48 h and transfected with different amounts of gRNA BisBG 180 for 24 h. For

biological replicates (*N* = 2) and more details, see Supplementary Fig. 2. **g** Time-dependent editing yield after single (T1), double (T2, 48 h post T1) and triple transfections (T3, 48 h post T2) with 2 pmol of BisBG guide RNA (BisBG 180). **h** Editing in human primary cells, after adenoviral delivery of SA1Q (25-75 MOI), with different guide RNA amounts (0.2–5 pmol of BisBG 180). MOI multiplicity of infection, RPE retinal pigment epithelium cells, NHA normal human astrocytes. All guide RNA amounts denote pmol/96-well with 150 µl total volume. Data in panel b-e, g and h is shown as the mean ± s.d. of *N* = 3 independent experiments, individual data points are shown as dots. Source data and full western blots are provided as a Source Data file.

codons, editing yields could be boosted up to twofold. Thus, we suggest to regularly apply the deoxyinosine substitution when targeting 5′-CAN codons. Editing yields of the respective improved guide RNA designs were included into Fig. 3b, with a different color (orange).

The dataset from Fig. 3b clearly demonstrates large effects of guide RNA and/or target mRNA structure on editing. Repeatedly, we observed largely varying editing yields when targeting different tyrosine phosphorylation sites (5′-UAY codons, Y = U, C) or different lysine modification sites (5′-AAR codons, R = A, G) on the same

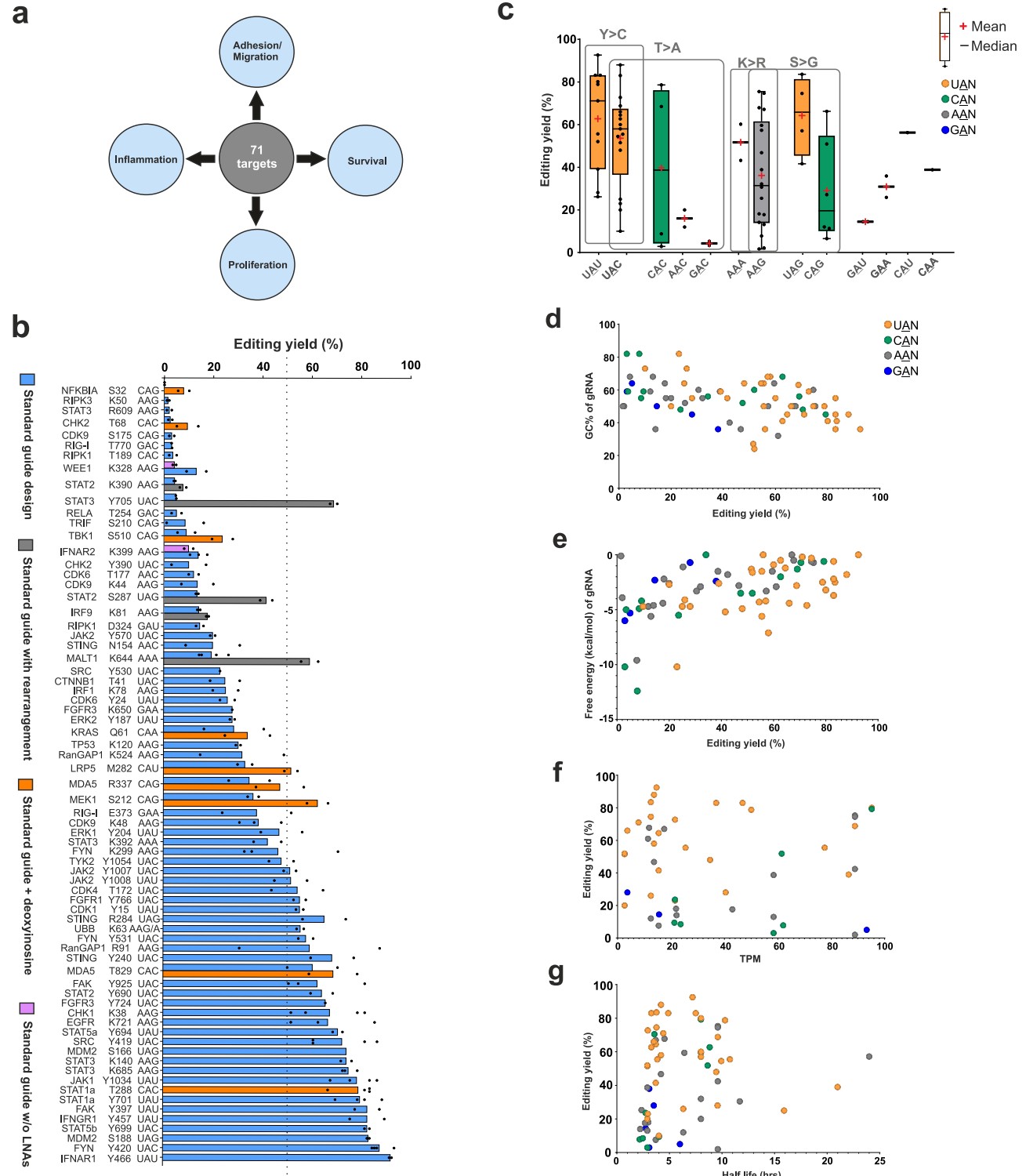

**Fig. 3 | Unbiased PTM interference screen with RNA base editing. a** Various signaling cues related to essential biological processes represent attractive targets for doseable and transient manipulation by RNA base editing. **b** 70 + PTM sites (Y > C, K > R, S > G, T > A, etc.) on various endogenous signaling transcripts have been targeted in HeLa-PB-SA1Q cells using 2 pmol (per 96-well with 150 µl total volume) of the improved standard guide RNA design (5′−3 + 9 + C + 12 nt, 4 LNAs, BisBG) or variation thereof as indicated. **c–g** Analysis of effects of various parameters on RNA base editing efficiency. Mean editing yields ($N = 2$) of 70+ targets plotted against target codon (**c**), GC content of the guide RNA calculated with Oligo Calc[44] (**d**), minimum free energy of guide RNA hairpin folding secondary

structure calculated with NUPACK[45] (**e**), relative gene expression level (mean TPM values of genes in Hela PB SA1Q cells with 48 h dox induced SA1Q expression (**f**). Half-life of the transcript, taken from[42]. In panels c-g, the color code indicates the nucleotide 5′ to the edited adenosine. Data in panel b is shown as the mean of $N = 2$ independent experiments in most cases and $N = 3$ or 4 in some cases, individual data points are given. In panel **c**, the box represents the interquartile range showing the middle 50% of the data points. The ends of upper or lower T-shaped whiskers extend to the maximum or minimum data point which is still within 1.5 times the interquartile range. Source data are provided as a Source Data file.

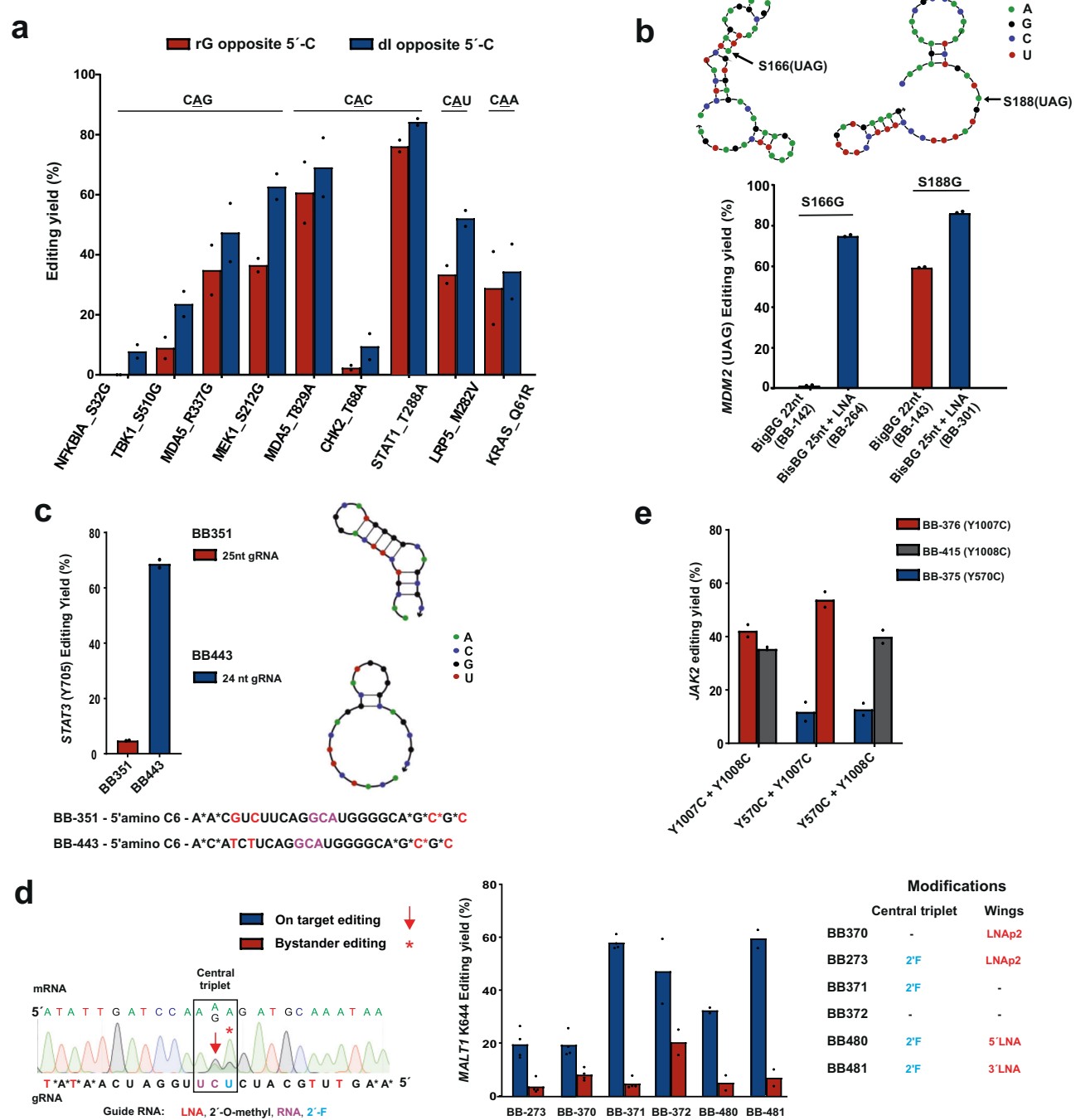

**Fig. 4 | Optimization of sequence and chemistry of guide RNAs for effiecient PTMi.** Comparison of different guide designs for improved editing of different signaling targets in HeLa-PB-SA1Q cells using 2 pmol (per 96-well with 150 μl total volume) of the respective BisBG-gRNA (BB-gRNA). For all gRNA sequences and modifications, see target list in the Supplementary Data 1. **a** Placement of deoxyinosine opposite the orphan cytidine in 5′-CAN (N = A, U, G, C) target codons did clearly foster editing efficiency, as shown for nine examples covering all four 5′-CAN codons. **b** LNA nucleotides were essential to achieve high editing efficiency in a structured mRNA substrate (*S166* in *MDM2*); arrows indicate the target adenosine in the structured *MDM2* mRNA. **c** Relaxing guide RNA hairpin structure by sequence optimization was essential for editing *Y705 > C* in *STAT3*.

**d** Placement of a 2′-F modification opposite the 3′-terminal adenosine base was required to suppress bystander editing in the adenosine-rich 5′-AAA codon. In this particular target, LNA nucleotides were only well accepted at the 3′-terminal half of the guide RNA. **e** Multiplexing RNA base editing by co-transfection of two guide RNAs against up to three different sites on the *JAK2* transcript (*Y570, Y1007, Y1008*). Two adjacent sites, e.g., *Y1007/Y1008*, can be targeted by two individual guide RNAs without compromising editing efficiency. Data in panel **a**–**c** and **e** is shown as the mean of *N* = 2 independent experiments, and in panel d, is shown as the mean of *N* = 2 (in most cases and in some cases *N* = 4), individual data points are given. Source data are provided as a Source Data file.

transcript side-by-side. On the *SRC* transcript, we found editing yields of >70% (Y419C) versus 23% (Y530C), on *JAK2*, we observed 50% (Y1007C, Y1008C) versus 20% (Y570C), and on *CDK9* we detected 40% (K48R) versus 13% (K44R). The most dramatic effect we observed was with the E3 ligase *MDM2*. With a prior guide RNA design (22 nt, no LNA,

but with BisBG), we found no detectable editing for the S166G site, whereas the S188G target site – only 66 nt downstream – gave editing with up to 60% yield. Both sites are highly editable 5′-UAG codons. By applying the mfold tool[46], we identified a strong local secondary structure at S166, but not at S188 (Fig. 4b). In contrast, both guide

RNAs did not contain noteworthy secondary structures. Thus, the different degrees of secondary structure of the two target sites seemed to cause the different editing yields. Importantly, both sites were well edited (to ca. 80%) when we switched to the improved design with 25 nt and four LNA bases, indicating that the increased length and the additional binding power provided by the LNA nucleotides help to make the guide RNA invade into this structured target site.

When we analyzed the editing of the *STAT3* transcript, we were wondering why we achieved excellent editing on two lysine PTM sites (ca. 75% for K140R and K685R), but why the more editable 5′-UAC codon of the important phospho-tyrosine Y705 showed almost no editing (ca. 5%). When we analyzed the secondary structure of the guide RNA with the NUPACK tool[45], we found that this guide RNA folds into a very stable tetra loop hairpin with a stem of eight base pairs, which engaged three of the four LNA nucleotides (Fig. 4c). To break the structure, we adapted the guide RNA sequence. Specifically, we shortened the antisense part of the guide RNA by one nucleotide from the 5′ end and included a different 3 nt 5′-overhang that did not engage into secondary structure formation. This 24 nt guide RNA (5′−3nt + 8-C-12 nt, with four LNAs) achieved editing levels up to 70%, similar to the phosphor-tyrosine sites of other STAT members.

Bystander editing — the editing within the guide RNA / mRNA duplex — is a severe engineering challenge in competing approaches that apply genetically encoded guide RNAs[8,47–49]. In contrast, the SNAP-ADAR tool blocks bystander editing outside the central base triplet by global 2′-*O*-methylation of the guide RNA[11]. However, in highly adenosine-rich codons, bystander editing of the nearest neighbor can occur[12]. In the PTMi screen bystander editing occasionally happened during K > R editing at the 3′ nearest neighbor in 5′-AAA(G) codons, in particular when a guanosine followed that 3′ adenosine. A typical example from the screen is the removal of the regulatory mono-ubiquitination site K644R in *MALT1*, where we initially obtained an on-target editing yield of 20% and a bystander editing of 8% (Fig. 4d). By incorporating a 2′-F modification opposite the bystander position, the off-target yield was reduced to below detection (<4%). Interestingly, we found that the K644R site did not benefit from the LNA modification. In fact, a guide RNA containing the 2′-F modification but not the four LNA nt achieved editing levels up to 58% with very little bystander editing (<5%). According to the NUPACK tool, the guide RNA did not fold into a very strong secondary structure. We tested if LNAs at one of the termini were better accepted than on the other and found that two LNAs at the 3′-terminus were very well accepted in contrast to two LNAs on the 5′-terminus. We speculate that LNA nucleotides can sometime also negatively interfere with editing, this might be more relevant for moderate-to-edit codons like 5′-AAA. In general, a single 2′-F nucleotide is generally recommended opposite the 3′-terminal adenosine in 5′-AAA codons.

Finally, we tested the best strategy to concurrently edit two adjacent sites on the same transcript. For this, we co-transfected two out of three guide RNAs into Hela-PB-SA1Q cells, which target three distinct tyrosine residues (Y580, Y1007, Y1008) on the *JAK2* transcript. In principle, one would expect the guide RNAs targeting Y1007 and Y1008 to mutually compete for target engagement. Nevertheless, the editing yields were similar for Y1007/Y1008 guide RNA co-transfection compared to other combinations, e.g., Y1007/Y570 or Y1008/Y570, where the guide RNAs are not expected to mutually compete (Fig. 4e). This suggests that guide RNAs act on the target RNA rather in a hit-and-run fashion than staying at the target for a long time. We tested this hypothesis by trying to inhibit an editing reaction by co-transfection of a BisBG-guideRNA with an excess of an editing-incompetent NH2-guide RNA of that same sequence. In agreement with our model, even a tenfold excess of NH2-guide RNA was hardly able to reduce the editing yield (Supplementary Fig. 3).

## Perturbation of the JAK/STAT signaling pathway by PTMi

To demonstrate the potential of PTMi to perturb signaling cues, we tested the manipulation of Interferon-α (IFN-α, type I IFN) or Interferon-γ (IFN-γ, type II IFN) induced response, which is mediated via the JAK/STAT pathway[25–27]. The canonical IFN-γ signaling results in the activation of the STAT1 transcription factor by phosphorylation of tyrosine 701, homodimerization, and nuclear translocation, which finally leads to the expression of ISGs carrying the GAS promoter sequence [29] (Fig. 5a). IFN-α, on the other hand, leads to formation of the ISGF3 complex containing pSTAT1, pSTAT2 and IRF9, which leads to the expression of ISGs carrying the ISRE promoter element[28]. Various functionally important PTM sites, in particular phosphorylation and acetylation sites, have been described for all members of the JAK/STAT pathway starting from the IFN receptors down to the transcription factors[50–52]. Figure 5a illustrates various PTM sites color-coded for the estimated effect of PTMi on ISG expression, being either activating (green) or inactivating (coral red). We looked particularly deeply into two well-known STAT1 mutations, which have been found in patients to be either a dominant LOF[34](Y701 > C) or a dominant GOF[35] (T288 > A). PTMi experiments were carried out in HeLa-PB-SA1Q cells. Such cells respond well to IFN-α or -γ treatment, resulting in pY701 STAT1 levels which are clearly detectable by Western blot (Fig. 5b, Supplementary Fig. 4). As expected, the response to IFN-γ gave higher pSTAT1 levels than IFN-α. We then studied the effect of introducing the respective LOF or GOF mutation by RNA base editing on the IFN response. When cells were transfected two times (T2), 72 h and 24 h prior to IFN treatment, with a guide RNA that introduces the LOF mutation Y701 > C (BB180, 80% editing yield), and lysed 24 h after IFN treatment, we found a clear reduction in pSTAT1 levels by Western blotting (0.5-fold of the unedited control for IFN-γ). In contrast, when cells were transfected with a guide RNA that introduces the GOF mutation T288 > A (BB478, 80%), we found a shallow increase in pSTAT1 levels (1.2-fold of the unedited control for IFN-γ), see Fig. 5b, Supplementary Fig. 4. Binding of the guide RNA or the guide RNA-SNAP-ADAR conjugate could potentially affect STAT1 levels negatively. However, neither an amino guide RNA control (incompetent to recruit SNAP-ADAR) nor an editing-incompetent but conjugation-competent control guide RNA negatively affected pSTAT1 levels, see Supplementary Fig. 4. We next characterized the effects of PTMi on the IFN-γ response by RT-qPCR and Immunofluorescence imaging. First, we measured the relative expression level of the well-known GAS-driven ISGs, *CXCL9* and *IRF1*. Indeed, both *CXCL9* and *IRF1* were strongly activated (ca. 9.000-fold and ca. 40-fold, respectively) by IFN-γ in the control sample lacking PTMi. However, after introducing the dominant LOF mutation Y701 > C via PTMi, the expression of both *CXCL9* and *IRF1* were strongly damped (ca. 10-fold against the unedited IFN-γ control). On the other hand, introducing the dominant GOF mutation T288 > A resulted in a moderate increase of *CXCL9* expression (1.5-fold against the unedited IFN-γ control) and a mild increase of *IRF1* expression (1.2-fold against the unedited IFN-γ control), see Fig. 5c. This indicates that PTMi via RNA base editing allows to manipulate the JAK/STAT signaling pathway up or down depending on the selected PTM site. Then, we further characterized the signaling event by immunofluorescence imaging against total STAT1. Prior to IFN-γ treatment, total STAT1 was mainly residing in the cytoplasm. Quickly after adding the cytokine (30 min), STAT1 almost entirely localized to the nucleoplasm. Around 8 h after IFN-γ treatment, STAT1 repopulated the cytoplasm (Fig. 5d, Supplementary Fig. 5). We repeated the experiment but introduced the respective LOF (Y701C) or GOF (T288A) mutation by transfecting the respective guide RNA (BB180 or BB478), 72 h and 24 h prior to IFN-γ addition and monitored the localization of total STAT1. As expected, the Y701C LOF mutation created a STAT1 variant that was impaired to move into the nucleus, in accordance with the strong damping of ISG expression. In contrast, the GOF T288A mutation created a STAT1 variant that went well into the

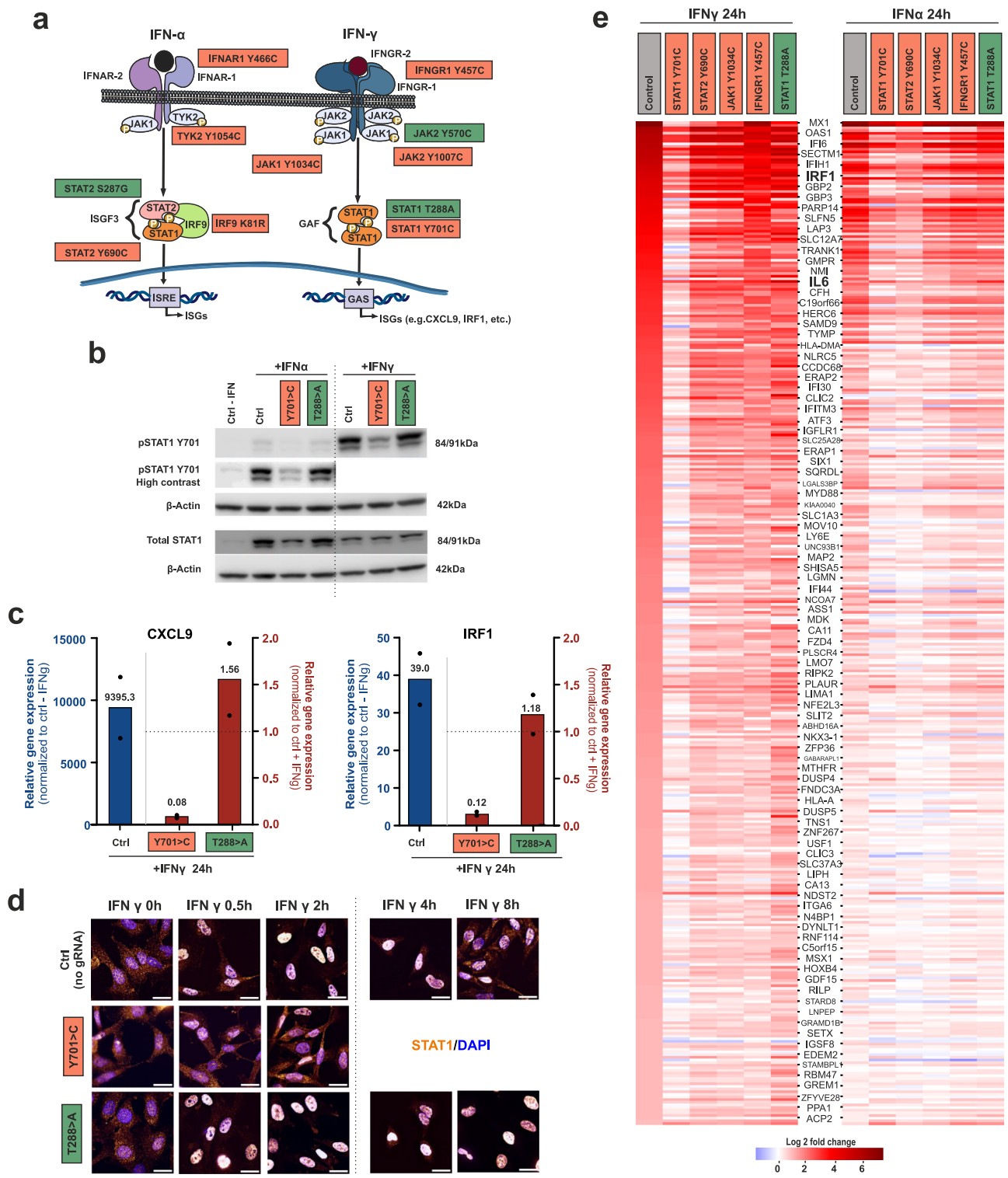

**a** IFN-α / IFN-γ signaling pathway schematic with PTM mutations: IFNAR1 Y466C, TYK2 Y1054C, STAT2 S287G, STAT2 Y690C, IRF9 K81R, IFNGR1 Y457C, JAK2 Y570G, JAK2 Y1007C, JAK1 Y1034C, STAT1 T288A, STAT1 Y701C.

**b** Western blot: pSTAT1 Y701, pSTAT1 Y701 High contrast, β-Actin, Total STAT1 with +IFNα and +IFNγ treatments.

**c** CXCL9 and IRF1 relative gene expression.

**d** STAT1/DAPI immunofluorescence at IFN γ 0h, 0.5h, 2h, 4h, 8h for Ctrl (no gRNA), Y701>C, T288>A.

**e** Heatmap of gene expression (Log 2 fold change) for IFNγ 24h and IFNα 24h across Control, STAT1 Y701C, STAT2 Y690C, JAK1 Y1034C, IFNGR1 Y457C, STAT1 T288A.

nucleus, but which stayed nuclear for prolonged time compared to the non-transfected IFN-γ-treated control, indicating a mechanism for increased IFN-γ signaling. Both results are well in accordance with the literature[34,35,53].

To get a broader picture of PTMi on the interferon-induced JAK/STAT pathway, we compared the editing of six different PTM sites at five different players of the pathway by double-transfection (T1 – 24 h after Doxycycline induction, T2 – 48 h post T1) of the respective guide RNA into HeLa-PB-SA1Q cells and studied global gene expression changes 24 h after IFN-α or -γ addition via next generation sequencing

(DESeq analysis[54]). Specifically, we individually introduced the following PTM mutations: Y701 > C in STAT1[34] (BB180, ca. 60% editing yield), T288 > A in STAT1[35] (BB478, ca. 80%), Y690 > C in STAT2[28] (BB349, ca. 62%), Y1034 > C in JAK1[55] (BB355, ca. 70%), Y466 > C in IFNAR1[56] (BB305, ca. 82%), Y457 > C in IFNGR1[57] (BB304, ca. 82%). Empty transfected cells with or without IFN treatment served as negative controls. By Western blot, we showed for all targets that the binding of the guide RNA does not negatively affect target protein expression (Supplementary Fig. 7). Poly(A) + RNA was collected 24 h after interferon treatment and global changes in transcriptome expression were compared to an unedited

**Fig. 5 | Modulation of the interferon α- and γ-response by PTMi. a** Scheme of the JAK/STAT pathway, in response to IFN-α/γ, initiating a phosphorylation cascade resulting in transcription factor activation, translocation and induction of an ISG response, figure adapted from[59]. Highlighted are specific PTMi target sites colored for the expected effect on the signaling outcome in coral red (downregulation) or green (upregulation), respectively. All experiments were performed in HeLa-PB-SA1Q cells, transfecting 2 pmol of the respective guide RNA 72 h and 24 h prior to IFN (2000 U/ml) stimulation for 24 h. **b** Western blot detection of STAT1 activation (via *Y701* phosphorylation) in response to either IFN-α or -γ, following on the introduction of either a GOF (*T288 > A*) or LOF mutation (*Y701 > C*) via RNA base editing. For full blots in duplicate, see Supplementary Fig. 4. **c** Analysis of the effect of PTMi (*STAT1*, LOF/GOF) on cytokine expression (*CXCL9* and *IRF1*) in response to IFN-γ stimulation by RT-qPCR, blue bars show expression levels in control with IFN-γ normalized to ctrl w/o IFN-γ; red bars show expression levels in samples with PTMi

and IFN-γ normalized to control with IFN-γ. **d** Analysis of subcellular localization of total STAT1 by Immunofluorescence imaging in response to IFN-γ treatment in control cells (IFN-γ for 0 h, 30 min, 2 h, 4 h, and 8 h) and in cells after PTMi of *STAT1* (LOF/GOF). Scale bars represent 25 μm. Images shown are zoomed-in, cropped and merged channels of STAT1 (amber) and nucleus (blue). For full and split channel images, see Supplementary Fig. 5. For statistical analysis of colocalization, see Supplementary Fig. 6. For detailed protocol, see Methods section. **e** Transcriptome-wide expression analysis (DESeq) on perturbing ISG expression in response to IFN-α or −γ treatment, via PTMi at five different JAK/STAT targets. In the control experiment (empty transfection), 379 ISGs were identified, which passed a significance threshold ($p < 0.001$) and a ≥1 log2-fold change in gene expression and were plotted for increasing induction in response to a 24 h IFN-γ stimulation. Data in panel c is shown as the mean ± s.d. of $N = 2$ independent experiments, individual data points are given. Source data are provided as a Source Data file.

control without interferon treatment. All experiments were carried out in duplicates. The DESeq2 pipeline[58] identified roughly 11,000 genes with TPM ≥ 2 and assigned $p_{adj}$ values to each transcript. From the control experiments (no PTMi) with versus without 24 h IFN-γ treatment, we selected 379 highly significantly ($p_{adj}$ <0.001) differentially expressed ISGs (≥1 log2 unit gene expression change), see Excel Sheet for heatmap in the Source Data. These ISGs were plotted in a heatmap format sorted for increasing response of the unedited control to IFN-γ stimulation, see Fig. 5e, heatmap at the left. Typical ISGs, like MX1 or IFI27, were strongly upregulated (>7 log2 units) in response to IFN-γ treatment. Notably, when introducing the Y701 > C LOF mutation in STAT1 preceding IFN-γ treatment, the ISG response was broadly and strongly inhibited, almost back to levels of control cells untreated with interferon. The editing of STAT2 or JAK1, in contrast, had comparably subtle effects on ISG expression, which is in accordance with their limited extent of contribution for activation of GAF complex. However, subtle differences in ISG gene expression were found indicating that unique ISG expression patterns are accessible by PTMi in these players. Interestingly, the editing of the IFN-γ receptor (IFNGR1) had particularly little effect on ISG expression, even though the editing level was very high (ca. 80%). This indicates that the selected mutation was not dominant, and that the remaining 20% unedited receptor might be sufficient to drive the full ISG response. Importantly, the editing of the GOF mutation T288 > A in STAT1 leads to a completely different, highly complex modulation of ISG expression, including the activation of numerous genes, including important cytokines like IL6 and CCL2 by ≥1.8 log2 units. Clearly, PTMi enables the damping, the modulation, but also activation of the IFN-γ signaling pathway depending on the PTM site selected. Upon IFN-α treatment, the changes in ISG expression in the unedited control where less strong compared to IFN-γ treatment (Fig. 5e, heatmap at the right). The removal of the pY701 site in STAT1 had again a damping effect on ISG expression. Interestingly, this was also true for the analog site (Y690 > C) in STAT2, highlighting a clear difference between the IFN-α- and IFN-γ-driven ISG response. The dependency of ISG expression on STAT2 phosporylation in response to IFN-α stimulation, however, seems to mirror very well the requirement for both pSTAT1 and pSTAT2 in the ISGF3 complex. The editing of the other three targets had comparably little effects on ISG expression. Even though the editing yield of the IFN-α receptor (IFNAR1) was high (80%), the chosen LOF mutation might not have been dominant enough to result in a clear damping of the pathway. In accordance with the literature, the GOF mutation T288 > A in STAT1 had almost no activating effect on IFN-α-dependent ISG activation[35].

Overall, the data highlights that PTM interference – as induced by RNA editing – can manipulate signaling pathways in a unique, activating and/or inactivating manner. It also shows that differences in the pathways, like dependency of IFN-α but not IFN-γ on phosphorylation of STAT2, are reflected in the perturbation of the ISG signaling outcome, indicating that RNA editing can be used as a tool to interrogate the regulatory role of specific PTM sites in complex signaling networks.

Importantly, all PTM sites were studied in the same cell line that had the SNAP-ADAR effector stably integrated simply by transfection with the different guide RNAs. The approach avoids the cumbersome creation of genetically modified cell lines for each PTM site in each signaling molecule and it also avoids artefacts from the overexpression of cDNAs of signaling molecules carrying specific PTM mutations.

## Discussion

RNA drugs have recently been very successful in the clinics[60–62], providing safe and efficient technologies for target gene knockdown. RNA base editing technology now promises to expand the scope of current RNA therapeutics[63,64]. While it is obvious that RNA base editing could be applied to repair disease-causing G-to-A and T-to-C point mutations[65], the largest value of the approach might lay in the precise and safe modulation of native protein function to modulate essential biological processes, e.g., in metabolism or signaling. While the reversible nature of RNA base editing is a disadvantage for the correction of inherited disease-causing mutations as it requires continuous re-dosing, the transient mode of action is a unique property that will enable the manipulation of essential biological processes in a dose-dependent and thus safe manner at the RNA level. Twelve out of the twenty canonical amino acids can be substituted to interfere with protein function. With respect to the manipulation of signaling networks, the removal of regulatory PTM sites, e.g., sites of phosphorylation and/or acetylation, seems highly feasible by means of RNA base editing, as all relevant amino acids that carry these PTMs (Y, S, T, K) can be defunctionalized[20].

Our dataset gives a broad and unbiased view on the accessibility of different PTM sites for RNA base editing over a whole range of well-known signaling proteins (Fig. 3). Besides the well-studied codon preference of ADAR[12,66], we identified the secondary structure of guide RNA and/or of the target RNA as a key factor that determines editing efficiency (Figs. 3 and 4). This is in accordance with previous studies focusing on the design process of ADAR-recruiting CLUSTER guide RNAs, where the *in-silico* optimization to reduce guide RNA secondary structure was critical too[49]. Furthermore, our data set predicts that the expression levels and half-lives of the target RNAs are less critical to achieve high editing yields. However, regarding the half-lives, this conclusion can only be drawn for half-lives ≥2 h due to limitations in the dataset that provided the degradation kinetics[42]. Thus, it may well be that very short-lived RNAs, like *NFKBIA*, were not well editable due to their very short life cycle[67].

RNA interference (RNAi) had remarkable clinical success in recent times[68,69] with the efficient knockdown of target RNAs, which often code for proteins with toxic gain-of-function mutations. In contrast to RNAi, PTM interference (PTMi), as we suggest here, enables to precisely manipulate a specific protein function without removing the protein itself. This is an important difference to RNAi since many proteins fulfill several independent functions, e.g., a catalytic protein may serve as a binding platform of a larger protein complex, or a

transcription factor might serve as a node that integrates signals from different cues. As exemplified with the STAT1 Y701 > C substitution, the signaling function of STAT1 can be efficiently switched off without removing the protein from the cell (Fig. 5). As STAT1 Y701C has a dominant negative effect on IFN-γ signaling[34], the presence of the edited protein helps to mediate the damping of ISG expression upon IFN-γ treatment so that $a \approx 50\%$ editing yield is already sufficient to interfere strongly with JAK/STAT signaling. Furthermore, PTMi enables to positively modulate protein function, as we exemplify with the STAT1 T288 > A editing, which kept the signaling competent pSTAT1 for a longer time in the nucleus and increased the expression of a certain set of ISGs.

For many signaling pathways, there is already a large body of knowledge accessible on potentially attractive sites for PTM interference (see also the Supplementary Data 1 – Targets list for the PTM sites chosen for Fig. 3). However, we have to better understand what sites are well editable and what editing efficiency is sufficient to achieve a useful down-stream effect. The unbiased PTMi of six different sites on five different players of the JAK/STAT pathway gave some first insights. While all PTMi experiments induced changes in the ISG expression pattern, the effect of the dominant-negative STAT1 Y701 > C was particularly strong, whereas the highly effective editing (≥80%) of the interferon-α and −γ receptors had only subtle effects on downstream signaling. Thus, PTMi to introduce either dominant negative loss-of-function or dominant positive gain-of-function mutations seem particularly effective to perturb a signaling pathway.

Engineered RNA base editing tools could be applied to screen for attractive PTMi targets in any given pathway. The results could then instruct the currently cumbersome engineering of RNA drugs that harness endogenous ADAR[47–49,70]. However, for this, we need reliable RNA base editing tools that apply simple guide RNA design rules and that achieve efficient and bystander-free editing. Currently, many RNA base editing tools have a restricted codon scope and difficulties to achieve bystander-free editing[16]. The improved SNAP-ADAR guide RNA design that we present here may largely serve this purpose. The SNAP-ADAR approach allows various chemical tweaks to improve editing efficiency and precision. We exemplify this by boosting the editing efficiency for the difficult-to-edit 5′-CAN codons (N = G, A, U, C) by site-specific incorporation of the non-canonical nucleoside inosine into the guide RNA (Fig. 4a). Furthermore, we could largely improve editing efficiency and potency by enforcing the dimerization[71,72] of SNAP-ADAR on the guide RNA (with the BisBG self-labeling moiety) and by incorporating up to four LNA bases. The latter was also shown to assist editing in structured target RNAs (Fig. 4b). Further, non-natural nucleoside variations have been reported to improve editing efficiency when used in the central base triplet (CBT), like 2′-fluoroarabinose[73], Benner's base Z[74] and nebularine[73], which might be well applicable in the SNAP-ADAR tool. Furthermore, the SNAP-ADAR tool achieves very high control of bystander editing, also in adenosine-rich transcripts, and even in adenosine-rich target codons, like 5′-AAA. Bystander editing outside the CBT is entirely suppressed by 2′-O-methylation of the SNAP-ADAR guide RNA. Inside the CBT, full control can be achieved by strategic placement of single 2′-F and/or 2′-O-methyl modifications[11,12] (Fig. 4d). This is particularly important for PTMi at lysine sites, where both target codons are A-rich (5′-AAR, R = G or A). Overall, the advantages of the SNAP-ADAR tool in terms of codon scope, efficiency, and bystander precision might compensate for the lack of encodability of the guide RNA, which makes screening set-ups more expensive compared to fully encodable approaches like the λN-ADAR[7,75] or Cas13-ADAR[9] approaches.

Overall, the data suggests that PTMi could become a major field of application of RNA base editing, in particular for clinical applications. Given that liver is an established target tissue for RNA drugs, we envision that PTMi could be used to modulate liver metabolism, e.g., to improve liver homeostasis in NASH or metabolic syndrome and to

develop therapies in the cardiovascular space. Furthermore, an efficient tool, like the SNAP-ADAR tool, might also allow to apply PTMi to probe for regulatory PTM sites, and thus to increase our understanding of the regulation of basic biological processes like signaling, metabolism, transcription, epigenetics, and protein degradation. In comparison to studying such processes with DNA (base) editing (with CRISPR/Cas tools), the fast (editing with the SNAP-ADAR tool achieves maximum editing yields within hours after transfection) and transient nature of RNA base editing is a key advantage to avoid problems arising from genetic compensation or lethality, in particular when essential processes are targeted[21,22]. PTMi with RNA base editing could also be applied to modulate the function of high-copy genes, like ubiquitin or histones, which are inaccessible to DNA editing tools, further highlighting the promises of the approach.

## Methods

### Guide RNA synthesis

Guide RNAs were purchased either from Biospring (Frankfurt, Germany) or Eurogentech (Seraing, Belgium) as HPLC-purified, chemically modified single-stranded RNAs carrying a 5′-C6 amino linker for subsequent installation of the self-labeling moiety. The sequences and chemical modifications of all guide RNAs are presented in Supplementary Data 1 (Target list sheet). The single BG or BisBG self-labeling moiety was synthesized as described recently[12,76] and was attached to the gRNAs using a slightly modified, recently reported improved protocol as follows[15]: BG/BisBG connected to a carboxylic acid linker (4 µl, 60 mM in DMSO) was activated in situ by incubation with DIC (3 µl, 270 mM in DMSO), DIPEA (2 µl, 5% in DMSO) and NHS (2 µl, 460 mM in DMSO) for 4 h at 45 °C with shaking at 900 rpm. This pre-activation mix was freeze-dried to remove excess DIC and re-suspended in 12 µl of DMSO/DIPEA (60/1). Then, the NH₂-guide RNA (4 µl, 10 µg/µl) was added to the pre-activation mix and incubated (2 h, 37 °C, 900 rpm).The crude BG/BisBG-linked gRNA was purified from unreacted NH₂-guide RNA by 20% urea PAGE and extracted with H₂O. RNA precipitation was done with (0.1 volumes, 3.0 M) and ethanol (3 volumes, 100%, overnight at −20 °C). The BG/BisBG-guide RNA pellet was washed with 75%Ethanol, air-dried, and dissolved in deionized water (25 µl). The concentration was determined by measuring absorbance at 260 nm.

### Cell culture

Flp-In T-REx 293 cells (cat. no. R78007, Thermo Fisher scientific) stably expressing SA1Q were cultured and used for editing experiments as previously described[12]. Ad293 cells (Cat. No. 240085, Agilent Technologies) were cultured in DMEM with 10% FBS and 1% Antibiotic-Antimycotic. A549 and HeLa cell lines (ACC 107 and ACC 57, DSMZ, Braunschweig, Germany) were cultured in DMEM with 10% FBS and 1% Penicillin/Streptomycin (P/S). Primary cells, NHA (Normal Human Astrocytes, Cat.no. CC-2565) and H-RPE (Human Retinal Pigment Epithelial Cells, Cat. no. 00194987), were purchased from Lonza and cultured in their respective commercially available medium and supplements from Lonza as instructed in the manual (AGM#- Astrocyte Growth Medium BulletKit, Cat. no. CC-3186 for NHAs and RtEGM#- Retinal Pigment Epithelial Cell Growth Medium BulletKit Cat. no. 00195409 for H-RPE cells). HEK 293FT-cells (Cat. No. R70007, ThermoFisher Scientific) were cultured in DMEM with 10% FBS and 500 µg/ml Geneticin. All cells were maintained at 37 °C with 5% CO2 in a water-saturated steam atmosphere. Authentication via STR profiling was performed by the commercial suppliers before purchase of the material. Cell lines were not additionally authenticated by us.

### Generation of SNAP-ADARs expressing cells

**PiggyBac HeLa and A459 cell lines.** To produce cells that stably express SA1 or SA1Q[12] on doxycycline induction, the previously described TRE3GS inducible PiggyBac plasmid Xlone was used[77]. XLone-GFP

plasmid was obtained from Addgene (Cat.No. 96930) and the ORF of the blasticidin resistance gene (Bsd) was replaced with a puromycin resistance gene (PuroR) by restriction cloning, resulting in the XLone_Puro_EGFP plasmid. Then GFP construct in both, Blasticidin- and Puromycin-resistant carrier plasmids was replaced either by SNAP-ADAR1 E406Q or wildtype SA1 construct by restriction cloning, resulting in the SA1Q_XL_BSD (pTS 1018) or SA1_XL_BSD (pTS 990); SA1Q_XL_Puro (pTS1086) or SA1_XL_Puro (pTS 1074) plasmids, respectively. For maps and sequences see Supplementary Information.

We then used the above-mentioned plasmids to generate stable transgenic HeLa and A549 cell lines. For this, $2\times10^5$ HeLa or A549 cells were seeded in 1 ml DMEM with 10% FBS in a 12-well plate. After 24 h, medium was replaced with 0.8 ml fresh DMEM with 10% FBS, and 200 μl of the transfection mix was added to each well (1600 ng of the respective PiggyBac carrier plasmid and 400 ng PB Transposase plasmid diluted in 100 μl OptiMEM and mixed with 6 μl Lipofectamine™ 3000 transfection reagent (Cat.no. L3000150, Thermofischer Scientific) diluted in 94 μl OptiMEM). After 24 h, the cells were scaled up to a 6-well plate. 48 h later, the medium was replaced with DMEM + 10% FBS + 1 μg/ml puromycin or 25 μg/ml blasticidin according to the PiggyBac plasmid transfected. After 12 days of selection, the medium was replaced by DMEM + 10% FBS + P/S. In this study, Hela cells were selected with Blasticidin and were used without further selection. A549 cells were selected with Puromycin and subsequently cultured sparsely in 96-well plate to allow for monoclonal selection. SA expression after doxycycline-induction was determined in A549 cells by fluorescence imaging after BG-FITC staining. Clones with strong and homogenous SA expression were selected and expanded for downstream experiments. HeLa-PB-SA1Q and A549-PB-SA1Q cell lines were maintained in DMEM + 10% FBS + P/S and validated by a guide RNA/BG-FITC-protein conjugation assay (Supplementary Fig. 2).

**Adenovirus production and transduction.** The SA1Q cassette was cloned into the pAdTrack-CMV backbone (Addgene, no. 16405, under CMVd1 and CMVd3 minimal promoters respectively pTS2396, for map and sequences see Supplementary Information. Pme-I (NEB, no. R0560S) linearized shuttle vectors containing the gene of interest were then delivered into BJ5183-AD-1 *E. coli* (Agilent, no. 200157) via electroporation with a Bio-Rad Genpulser at 1.6 kV, 200 Ohm and 25 μF. Plasmids were isolated from bacterial cultures using a Gravity-Flow Plasmid Mini-Kit (Qiagen, no. 12123). Recombinant adenoviral plasmids were verified as containing the gene of interest by Pac-I (NEB, no. R0547L) control digestion. For larger-scale production of recombined plasmids, these were retransformed and isolated from bacterial cultures using a Gravity-Flow Plasmid Midi-Kit (Qiagen, no. 12143). For virus production, 30–100 μg of adenoviral production plasmid was digested with Pac-I and purified by ethanol precipitation; 10 μg of the digested plasmid was then transfected into 40–80% confluent Ad293 cells (15-cm plate) using a plasmid: Lipofectamine-2000 ratio of 1:3. Within 7–10 days the emerging widespread cytopathic effect indicated successful adenovirus production. Cells were then harvested, pelletized, and treated with three freeze–thaw cycles at −80 °C in 1 ml of PBS to release adenovirus. Lysates were cleared from cell debris by centrifugation (10 min, 700×g). After the addition of 10% glycerol to the supernatant, viruses were aliquoted and stored for further use at −80 °C. Titer was determined using the Adeno-X Rapid Titer Kit (CloneTech/Takara, no. 632250) performed according to the manufacturer's protocol. Purified viruses were tested for E1A negativity, and thus replication deficiency, by immunoblot (Mouse Anti-Adenovirus Type 5 E1A, BD Pharmingen, no. 554155).

For transduction $5 \times 10^4$ A549 or Hela cells were seeded per 24-well in 500 μl medium. 24 h later, medium was exchanged to 400 μl and transduced with the respective amount of virus (25-75 MOI) diluted in 100 μl 1xPBS per well. 48 h post-transduction, cells were washed thrice with 1xPBS, detached, and used for further editing experiments (explained later) by reverse transfection of the guide RNA using

Lipofectamine™ 2000 transfection reagent (Cat. Number: 11668019). In case of primary cells, $2*10^4$ NHAs or $3*10^4$ RPEs were seeded per 96-well plate in 100 μl medium. 24 h later, medium was exchanged to 80 μl and transduced with the respective amount of virus diluted in 20 μl 1xPBS per well. 48 h post-transduction, cells were washed thrice with 1xPBS but were forward transfected with guide RNA using Lipofectamine™ RNAiMAX (Cat. number: 13778075, Thermofischer Scientific) in the same plate for further editing experiments.

**Lentivirus production and transduction.** The CMV promoter of the pLenti-Puro lentiviral vector plasmid (Addgene, Cat.no. 39481) was replaced with an EF1a promoter. N-terminally GFP-tagged SNAP-ADAR constructs (SA1wt or SA1Q) were cloned into the MCS of the plasmid to produce plasmids pTS1036 and pTS 1025, see Supplementary Information for maps and sequences. For virus production, $3*10^6$ HEK 293FT cells were plated on 10 cm (Ø) dish in DMEM + 2% FBS medium. After 24 h, medium was changed to OptiMEM and the cells, with 50-70 % confluency, were transfected with 3 μg of lentiviral vector plasmid along with 3 μg of ViraPower lentiviral Packaging Mix (Cat.no. K497500, ThermoFisher Scientific) using Lipofectamine™ 2000 transfection reagent (Cat. Number: 11668019) according to the manufacturer's protocol. The medium was changed 24 h after transfection to DMEM + 2% FBS. 48 h later, medium was collected, centrifuged to pellet debris, and the supernatant was filtered through a 45 μm PVDF filter (Carl Roth). Viral particles were further concentrated ca. 100-fold by centrifugation in 20 ml Vivaspin tubes (Sartorius) and aliquots were stored at −80 °C. The lentiviral stocks were titered by counting the transduced colonies after transduction in 6-well plates with 10-fold serial dilutions.

For transduction, $6\times10^4$ Hela or A549 cells were seeded in a 24-well with 300 μl of DMEM + 2% FBS medium, and viral particles were added at a MOI of 5 ($1.4*10^5$ TU/ml). After 24 h, DMEM + 10% FBS medium was added. 4 –7 days post-transduction, 10% of the cells with highest GFP expression were selected by FACS for further use. For editing experiments, cells were reverse transfected with gRNA as explained below (section Editing experiments).

**RNA editing**

**Single transfection.** Editing experiments in Flp-In-T-Rex 293 cells stably expressing SNAP-ADAR were done as previously described[12]. For HeLa and A549 PB cell lines, the procedure slightly differed in the number of cells and amount of doxycycline used: $2\times10^5$ cells were seeded in 24-well plates for SA expression by 1 μg/ml of doxycycline induction for 24 h. Later, cells were detached and $5\times104$ cells per 96-well were re-suspended in 100 μl DMEM with 10% FBS and 1.5 μg/ml Doxycycline and reverse transfected with the respective amount of guide RNA (as indicated for each experiment) mixed with 0.75 μl Lipofectamine 2000 in 50 μl OptiMEM. For the concurrent editing experiments, required guide RNA amounts of the respective guide RNAs (in this study, two guide RNAs) were mixed with 0.75 μl lipofectamine 2000 in 50 μl OptiMEM for reverse transfection. Lentivirally transduced cells constituently expressed SNAP-ADAR and so were directly reverse transfected with the indicated amount of guide RNA mixed with 0.75 μl Lipofectamine 2000 in 50 μl OptiMEM. For adenovirally transduced HeLa or A549 cells (48 h post transduction), cells from 24-well plate were washed, detached and $5\times104$ cells per 96-well were re-suspended in 100 μl DMEM with 10% FBS and reverse transfected with the indicated amount of guide RNA mixed with 0.5 μl RNAiMax in 50 μl OptiMEM. For adenovirally transduced primary cells, cells seeded in 96-well plates were washed 48 h post transduction, and 100 μl of respective fresh medium was added. Cells were then forward transfected with the indicated amount of guide RNA mixed with 0.5 μl RNAiMax in 50 μl OptiMEM.

**Multiple transfections.** For editing experiments with guide RNA transfections two or more times $2x10^5$ Hela PB SA1Q cells were seeded

in 24-well plates for SA expression by 1 μg/ml of doxycycline induction for 24 h. Later, cells were detached and $5\times10^4$ cells per 96-well were re-suspended in 100 μl DMEM with 10% FBS and 1.5 μg/ml Doxycyline and reverse transfected once with 2 pmol of respective guide RNA mixed with 0.5 μl RNAiMax in 50 μl OptiMEM. 24 h following each reverse transfection, cells were re-seeded in 24-well plates in DMEM with 10% FBS and 1 μg/ml Doxycycline. A second or third transfection was done 48 h post the respective previous transfection. 24 h after the last guide RNA transfection, depending on the experiments, cells were either treated with or without IFNα/γ (2000U/ml) for the indicated time points and then either lysed for RNA/protein isolation or fixed for imaging.

**RNA isolation and analysis of editing yields.** Cells were harvested in RLT buffer (Cat. No. 79216, Qiagen). Total RNA was extracted with the Monarch® RNA Cleanup Kit (New England BioLabs) following manufacturer's instructions, followed by DNaseI digestion. Then the residual guide RNA was sequestered by a complementary DNA oligo (for sequences see Supplementary Data 3), and RNA was converted into cDNA for subsequent amplification by either the One Step RT-PCR Kit (BiotechRabbit) or the OneTaq® One-Step RT-PCR Kit (New England Biolabs) with the appropriate primers. The DNA was analyzed by Sanger sequencing (Eurofins Genomics or Microsynth). A-to-I editing yields were quantified from the Sanger trace by measuring the height of the resulting guanosine peak divided by the sum of the peak heights of the guanosine and adenosine peaks at a respective site.

**Biochemical assays**

**Guide RNA/BG-FITC-protein conjugation assay.** Cells were seeded on a 6-well plate in DMEM + 10% FBS with Doxycycline (Dox – 1 μg/ml for HeLA/A549-PB-SA1Q cells, 10 ng/ml for Flp-In-Trex- SA1Q cells), 8 or 24 h prior to gRNA transfection. Later, cells were transfected with varying amounts of guide RNA as mentioned for each experiment. Cells with full induction of SA1Q (48 h Dox) had Dox added to the medium during transfection. 24 h later, medium was exchanged to DMEM + 10% FBS + O-acetylated BG-FITC (5 μM final concentration) and incubated for 30 min at 37 °C. BG-FITC stains unconjugated SNAP-ADAR. Cells were washed with 1xPBS and lysed using 100 μl of 1xLämmli buffer in Pierce RIPA lysis buffer (89900, Thermo Scientific) including complete protease inhibitor cocktail and PhosSTOP phosphatase inhibitor (4906837001 and 4693159001, Sigma). Lysate was flash-frozen in liquid nitrogen and stored at −80 °C. 20 μl of lysate was separated on 8-16% tris-glycine gels (Invitrogen) at 120 V for ~1.5 h in 1x SDS-PAGE buffer (0.25 M Tris, 1.92 M Glycin, 1% w/v SDS) and BG-FITC-stained, unconjugated SNAP-ADAR was visualized in the gel with a FLA 5100 Fluorescence Image Analyzer at 473 nm excitation. The Blot could be washed and further used for Western Blotting.

**Western Blotting.** Cells were harvested and lysed with Pierce RIPA lysis buffer (89900, Thermo Scientific) including complete protease inhibitor cocktail and PhosSTOP phosphatase inhibitor (4906837001 and 4693159001, Sigma). 30 μg of Protein lysates were separated on 8-16% tris-glycine gel (Invitrogen) and then blotted onto PVDF membrane (Cat. no. 88520, Thermo Scientific). Blocking was done with 5% dry milk powder in TBST buffer for 1 hr at room temperature, then washed three times with TBST. Membranes were then incubated overnight with primary antibodies diluted in 5% dry milk in TBST at 4 °C. Later, the blot was incubated with the respective HRP-conjugated secondary antibodies (rabbit 111-035-003; mouse 115-035-003; Jackson ImmunoResearch Laboratories) diluted 1:10,000 in 5% dry milk powder in 1x TBST and incubated for 2 h at r.t. on the membranes. Later, membranes were washed to prepare for imaging with 1x ECL solution plus 0.03% $H_2O_2$. Chemiluminescence was recorded with an Odyssey Fc Imaging System (LiCor Bioscience) and images were processed with

Fiji ImageJ. A full list of primary and secondary antibodies can be found in the Supplementary Table 1.

**Immunofluorescence microscopy.** Cells on coverslips were fixed with 3.7% formaldehyde for 20 min at RT., washed 3x with 1x PBS, and permeabilized with 300 μl ice-cold methanol for 10 min at −20 °C. After blocking in 10% FBS in 1x PBS overnight at 4 °C, samples were incubated overnight with primary antibodies diluted in 1x PBS, 5% FBS at 4 °C. Samples were then washed in 1x PBS and incubated in corresponding secondary IgG antibodies coupled to an Alexa Fluor dye (1:1000, Cell Signaling) for 1 h at RT. For a full list of antibodies, see Supplementary Table 1. Nuclei were stained with 1:200 NucBlue (Invitrogen) for 30 min at RT protected from light and again washed in PBS. Coverslips were then mounted using Fluorescent Mounting Medium (DAKO). Images were taken with a Nikon Eclipse Ti2-E inverted fluorescent microscope, equipped with a photometrics® Prime 95B camera and a lumencor® AuraII light engine. All pictures were recorded using a 60X oil objective (numerical aperture 1.4) and Olympus IMMOIL-F30CC immersion oil. The excitation wavelengths and corresponding filter sets used to record each channel are specified in Supplementary Table 2. A z-stack covering 6 μm (0.2 μm steps) was recorded. The same acquisition settings were chosen for each channel for all images. The images were deconvoluted using automatic 3D Deconvolution in Nikon NIS-Elements software and a single layer (z resolution -0.6 μM) is displayed. Further, assignment of lookup tables, maximum intensity projection, contrast settings and cropping were performed in FIJI ImageJ[78]. Amber LUT, used to display STAT1 protein, was obtained from ImageJ Wiki – NucMed LUT list (https://imagej.net/ij/download/luts/NucMed_Image_LUTs). Quantitative map between the LUT and the bitmap are provided in Source data file. For statistical analysis of colocalization, an unbiased sectioning of biological replicate images (N = 2) into four quadrants was done to determine Mander's coefficient with Costes' automatic threshold using the JACoP plugin[79] in ImageJ[78] for each condition. Significance was determined by an Unpaired, two-tailed $t$-test.

**Quantitative real-time PCR.** Following the respective gRNA double transfection and subsequent IFN treatment, cells were lysed in 300 μl Lysis buffer from a MonarchTotal RNA Miniprep Kit (NEB) and RNA was isolated following the manufacturer's protocol, including an on-column DNase digestion. cDNA was prepared using the High-Capacity cDNA reverse transcription kit (Applied Biosystems). Synthesised cDNA was purified with a NucleoSpin Gel and PCR CleanUp kit (Macherey-Nagel) and eluted with 20 μl nuclease-free water. The yield was determined using a Spark Microplate reader measuring absorption at 260 nm and dilutions of 10 ng cDNA/ml were prepared. Quantitative PCR (qPCR) runs were conducted in Fast 96-well plates (Cat. No. 4346907, Applied Biosystems) using Fast SYBR Green Master Mix (Cat. no. 4385612, Applied Biosystems) according to manufacturer's protocol (10 μl Fast SYBR Green Master Mix, 7.2 μl of nuclease-free water and 0.4 μl of each primer (10 μM) primer plus 2 μl of 10 ng/μl cDNA template or nuclease-free water for negative control) in an Applied Biosystems 7500 qPCR cycler (40 cycles-3s 95 °C, 30s 56 °C). Samples were measured in two technical replicates. GAPDH and ACTB were used as house-keeping genes (HKGs). A baseline correction was performed for each dataset using the 7500 data analysis. C(t) values were determined with a threshold value of 0.2. Fold change was calculated by ΔΔC(t)-method using the Geometric mean of the HKGs. The primer list can be found in Supplementary Data 2.

**Gene expression analysis of PTMi via NGS (DESeq).** $4\times10^5$ Hela PB SA1Q cells were seeded in 6-well plates for inducing SA1Q expression with 1 μg/ml of doxycycline for 24 h. Then, cells were detached and for each 24-well, $2\times10^5$ cells were re-suspended in 400 μl DMEM with 10% FBS and 1.5 μg/ml doxycyline and reverse-transfected with 8

pmol of the respective guide RNA after mixing with 2 µl RNAiMax in 200 µl OptiMEM. 24 h following the first transfection, cells were re-seeded in 12-well plates in DMEM with 10% FBS and 1 µg/ml doxycycline. 48 h post the first transfection, a second transfection was done similarly. 24 h following the second transfection, cells were re-seeded in 12-well plates in DMEM with 10% FBS, 1 µg/ml doxycycline, and induced with either IFN-α or IFN-γ (2000U/ml) for 24 h. Negative control samples were obtained by (1) empty transfection after doxycycline induction of SA1Q without IFN-α or IFN-γ induction, (2) empty transfection after doxycycline induction of SA1Q with IFN-α or IFN-γ induction. 24 h post IFN-α or IFN-γ induction, cells were lysed, and RNA was isolated using the Monarch® Total RNA Minprep (New England BioLabs) following manufacturer's instructions. 25 µl of purified RNA ($\geq 40$ ng/µl) was delivered to CeGaT (Germany) for poly(A)+ mRNA sequencing. The library was prepared from 200 ng of RNA with the TruSeq stranded mRNA library prep kit (Illumina) and sequenced with a NovaSeq 6000 (25 million reads, 2 × 100 bp paired end; Illumina). Each sample was prepared as biological duplicate. We used STAR (v. 2.4.2a)[80] to align RNA-seq reads to the hg19 reference genome and ran RSEM (v. 1.2.21)[81] on the alignments to calculate read counts and TPM values. Using read counts, we analyzed gene expression for all genes with TPM $\geq 2$ (for both replicates) with the R package DESeq2[58]. Significantly expressed genes were defined by Padj<0.01 and |log2 fold change | =2.

**Data analysis.** Non-NGS data were analyzed using Excel 2016 and GraphPad Prism8. Figures were created with CorelDraw 2017 and BioRender. Heatmap was made using Jupyter Notebook (6.4.12)[82]. The manuscript was written with Word 2016. NGS data was analyzed using STAR (v. 2.4.2a) to align RNA-seq reads to the hg19 reference genome and ran RSEM (v. 1.2.21) on the alignments to calculate read counts and TPM values. Using read counts, gene expression was analyzed for all genes with TPM $\geq 2$ (for both replicates) with the R package DESeq2. qPCR analysis was performed using the 7500 data analysis software v2.3. Microscopy images were acquired and deconvoluted using NIS element software (version 14.0.0.0) and pseudo colored and further processed using ImageJ (1.54 f). Colocalization analysis was made using JACoP plugin[79] in ImageJ[78]. All the gray value calculation for western blots was done using ImageJ (1.53q). Guide RNA hairpin folding secondary structure calculated with NUPACK online tool. Oligo Calc: Oligonucleotide Properties Calculator, online tool was utilized for calculating melting temperatures of primers. Sanger sequence traces were analyzed using SNAP-Gene (version 4.2.11).

**Statistics and reproducibility.** No statistical method was used to predetermine sample size. No data were excluded from the analyses. The experiments were not randomized. The Investigators were not blinded to allocation during experiments and outcome assessment. The exact sample size and the statistical test are described in the Figure legends, with exact *p* values given in source data. The test used was a two-tailed Student's *t* test by Graphpad prism software 8.

### Reporting summary
Further information on research design is available in the Nature Portfolio Reporting Summary linked to this article.

## Data availability
The raw DESeq data generated in this study have been deposited in the NCBI GEO server under the accession code GSE264114. Source data are provided with this paper.

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

## Acknowledgements
We thank the University of Tübingen, the Deutsche Forschungsgemeinschaft (STA 1053/3-1, STA 1053/3-2, STA 1053/7-1, STA 1053/10-1, STA 1053/11-1, all granted to T.S.), the European Research Council (CoG no. 647328, PoC no. 101069246, both granted to T.S.), the VolkswagenStiftung (grant no. 96 876 to T.S.), German National Academy of Sciences Leopoldina (grant no. LPDS 2019-06 to P.V.) for generous support. We thank the group of Prof. Jennifer Ewald for assistance with fluorescence microscopy.

## Author contributions
K.K. and T.S. conceived the study. K.K., S.S., S.M.S., C.F., A.B., V.M., L.F., and J.J.B. did the wet lab experiments. A.H. prepared the Piggy Bac cells. J.W. and P.R. prepared Adenovirus and Y.F. prepared Lentivirus and the Lentiviral-transduced cells. P.V. and J.B.L. helped with the analysis and interpretation of NGS data. K.K., S.S., and T.S. prepared the manuscript. All authors proof-read the manuscript.

## Funding

## Competing interests
T.S., J.B.L., and P.V. are cofounders and shareholders of AIRNA Bio. P.V. is an employee at AIRNA Bio and AIRNA Bio Germany GmbH. T.S. and J.B.L are consultants to AIRNA Bio. The remaining authors declare no competing interests.
