## [Peer Review File · Nature Communications]

REVIEWER COMMENTS

Reviewer #1 (Remarks to the Author):

RNA base editing can perturb protein function at post transcriptional levels and if achieved with high efficiency and accuracy, it can enable instant, partial and reversible manipulation of key amino acid residues in signaling molecules. This mechanism can be used to study roles of post-translational modifications (PTM). While RNA base editing has unique advantage over other techniques which are designed to eliminate the entire target proteins (i.e., CRISPR, siRNA), RNA base editing has its own technical challenges that limit its applicability, efficacy, and accuracy. This study tackled those challenges and identified multiple key modifications that collectively improved overall RNA base editing yield across various types of codons including hard-to-edit ones. At the end, the authors applied the updated technique to modify various molecules in the JAK-STAT pathway at their key PTM residues (PTM interference) and evaluated the global outcome by mRNA-seq. The results revealed differential magnitude of impact on ISG expression following targeted PTMi on receptors vs kinases vs transcription factors. Also at individual gene levels, a “variegated” deviation from normal induction profile was observed with each PTMi.

The work offers important technical advances to improve the performance of RNA base editing. The results are supported by well controlled rigorous experiments. The trial to edit the JAK-STAT pathway at various steps/molecules successfully confirmed some of expected outcomes (strong LOF phenotype of STAT1-Y701C) but diverged in most other situations, revealing the subtlety and complexity of perturbing the pathway at different steps in the cascade. Overall, the manuscript will help advance the field not only technically but also conceptually on the idea of PTMi. While there is still a room for further technical improvement, the manuscript convincingly shows that RNA editing is a viable and unique approach to study signaling and beyond.

The below is the list of specific questions/comments that are all minor and related to JAK-STAT.

1. Fig. 5d (image): While representative images shown seem to support the authors' claim, some sort of quantitative evaluation will strengthen the argument. For example, a blinded investigator could score cells for nuclear v cytoplasmic signals, calculate a mean score per condition, and compare conditions with stats to prove stronger and prolonged nuclear signals for GOF.
2. Fig. 5e (heatmap): While global impact on the selected ISG expression is well presented, subtle but distinct impact on individual genes is hard to decipher. Further classification of ISGs into subsets/modules that are uniquely up/down regulated by each PTMi may be helpful.
3. STAT1 AD GOF phenotype (introduction p3): there are references published after ref34 to describe broader spectrum of STAT1 GOF phenotype that include impaired anti-viral defense (Toubiana et al Blood 2016). It might be helpful to cite it and modify the statement regarding anti-viral defense.

4. STAT1 GAF complex acting on a GAS promoter sequence (introduction p3): Reports with ChIP-seq/ATAC seq point to STAT binding to intronic and intergenic enhancers and consequence of STAT1 LOF on enhancers (i.e., Lau et al Nat Immunol 2018 – Fig.3). Therefore, a more updated view on ISRE and GAS motifs would include enhancers (not just promoters).

Reviewer #2 (Remarks to the Author):

The manuscript titled "An Improved SNAP-ADAR Tool Enables Efficient RNA Base Editing to Interfere with Post-translational Protein Modification" by Kiran Kumar and colleagues is a groundbreaking study that employs RNA editing by ADAR tool for manipulating post-transcriptional modifications. The authors present an improved methodology (mainly oligo design) for this established platform that yields impressive improvements in editing efficacy as well as for the understanding of the fundamental aspects of the editing process by pointing out the validity of the "hit and run" model.

However, the main focus of the work and its importance lie in the utilization of this exciting technology for editing phosphorylation sites and demonstrating its ability to do so in dozens of sites (over 70). This is an incredibly remarkable story, as the application of RNA editing allows for more relevant PTM in comparison to current approaches, mainly due to its ability to regulate in a dose-dependent and reversible manner, which is a clear advantage over DNA editing approaches.

I find this work comprehensive, rigorous, and well-written. I believe this approach will be utilized by a large number of groups around the world and revolutionize the way we study PTMs. It is clear that works on other PTMs are on their way as well.

In summary, this paper is a groundbreaking contribution to the fields of RNA editing and PTM. One minor suggestion is to add a paragraph about the possible therapeutic benefits of this approach of PTM editing, as it is clearly the next step.

Reviewer #3 (Remarks to the Author):

Here the authors describe improvements to their previously reported SNAP-ADAR system for directed RNA editing. This tool uses a SNAP-ADAR fusion protein and a benzylguanine-linked antisense

oligonucleotide guide to direct the ADAR to the target site and form the requisite duplex for editing. In this manuscript, the authors report optimization of the antisense oligonucleotide (e.g. length, incorporation of modifications like LNA, inosine, 2'-F nucleosides and use of a bis-benzylguanine to recruit an ADAR dimer, etc) and they test different methods for delivering the SNAP-ADAR fusion protein. With the improved guide design in hand, they then apply their directed editing tool to modify codons at post translation modification sites in various signaling transcripts in HeLa cells expressing a hyperactive mutant of the SNAP-ADAR demonstrating varying editing efficiencies depending on target with some very high levels (>90%) and some very low levels (<10%). Finally, they inhibit specific PTM sites in components of the JAK-STAT signaling pathway and evaluate the impact on gene expression in response to interferon stimulation. Directed RNA editing is becoming an important method for manipulating transcriptomes with significant implications for new therapeutics. This work represents an important advance as it illustrates how improvements can be made in RNA editing tools like the SNAP-ADARs and describes a novel application of the technology (control of post translational modifications). I do have the following comments, however.

1) The authors make reference to the effect of RNA editing being fast, at one point even saying the effect is “instant”, yet they show it takes 4-7 days to determine an editing phenotype at the protein level. This is not particularly fast, and it makes one wonder if this approach is best suited for probing signaling pathways mediated by much faster phosphorylation and dephosphorylations. A brief comparison to DNA editing is made at the end of the paper. However, the benefit of this approach over using, for instance, the DNA base editors developed by David Liu in a cell line like HeLa, is not clear. Would the timing of the experiment really be much different? More discussion on this comparison is justified. A potential key advantage of the use of this approach over DNA editing is the reversibility of RNA editing, but reversibility was not demonstrated in this paper.

2) Are the gene expression effects only due to editing of the codon or could RNP assembly on the transcript be responsible for (at least some) the effects observed? Along these lines, it seems imperative for an approach like this to establish that the SNAP-ADAR guided to the transcript by a high affinity oligonucleotide is not causing changes in protein expression. The authors provide a single Western blot analysis for the effect on STAT1 expression in the presence of their Y701C and T288A guides. However, there was no attempt to quantify the effects or to show reproducibility with replicate experiments. Furthermore, the authors do not show how the other guides analyzed in Figure 5e (STAT2 Y690C, JAK1 Y1034C or IFNGR1 Y457C) effect the expression levels of those proteins. For the gene expression changes to be ascribed only to changes in PTM of a particular protein, it must be established that the SNAP-ADAR + guide targeting a specific transcript does not alter expression levels for that protein. This should be done in a quantitative manner with replicates and statistical analysis.

Point-by-point Rebuttal Letter

Reviewer #1 (Remarks to the Author):

RNA base editing can perturb protein function at post transcriptional levels and if achieved with high efficiency and accuracy, it can enable instant, partial and reversible manipulation of key amino acid residues in signaling molecules. This mechanism can be used to study the roles of post-translational modifications (PTM). While RNA base editing has unique advantage over other techniques which are designed to eliminate the entire target proteins (i.e., CRISPR, siRNA), RNA base editing has its own technical challenges that limit its applicability, efficacy, and accuracy. This study tackled those challenges and identified multiple key modifications that collectively improved overall RNA base editing yield across various types of codons including hard-to-edit ones. At the end, the authors applied the updated technique to modify various molecules in the JAK-STAT pathway at their key PTM residues (PTM interference) and evaluated the global outcome by mRNA-seq. The results revealed differential magnitude of impact on ISG expression following targeted PTMi on receptors vs kinases vs transcription factors. Also at individual gene levels, a “variegated” deviation from normal induction profile was observed with each PTMi.

The work offers important technical advances to improve the performance of RNA base editing. The results are supported by well controlled rigorous experiments. The trial to edit the JAK-STAT pathway at various steps/molecules successfully confirmed some of expected outcomes (strong LOF phenotype of STAT1-Y701C) but diverged in most other situations, revealing the subtlety and complexity of perturbing the pathway at different steps in the cascade. Overall, the manuscript will help advance the field not only technically but also conceptually on the idea of PTMi. While there is still a room for further technical improvement, the manuscript convincingly shows that RNA editing is a viable and unique approach to study signaling and beyond.

RESPONSE: We thank the reviewer for this very positive and supportive comment on our achievements regarding the RNA base editing tool itself but also regarding the conceptual impact of the work for the field.

The below is the list of specific questions/comments that are all minor and related to JAK-STAT.

1. Fig. 5d (image): While representative images shown seem to support the authors’ claim, some sort of quantitative evaluation will strengthen the argument. For example, a blinded investigator could score cells for nuclear v cytoplasmic signals, calculate a mean score per condition, and compare conditions with stats to prove stronger and prolonged nuclear signals for GOF.

RESPONSE: In response to the reviewer’s suggestion, we have now analyzed the co-localization with an unbiased ImageJ plug-in to determine the Mander’s coefficient of total STAT1 protein signal with nuclear DAPI staining, see new Supplementary Figure S7. After Y701>C editing, colocalization is reduced with statistical significance ($P \leq 0.001$). After T288>A editing, the colocalization is increased at the late time point (8h) with statistical significance ($P \leq 0.001$).

2. Fig. 5e (heatmap): While global impact on the selected ISG expression is well presented, subtle but distinct impact on individual genes is hard to decipher. Further classification of ISGs into subsets/modules that are uniquely up/down regulated by each PTMi may be helpful.

RESPONSE: We thank the reviewer for this hint. We provide an excel sheet with all 380 ISGs that are shown in main Figure 5 panel e. The expert reader can have a deeper insight into changes in those ISGs and define subgroups/modules upon the different PTMi target sites. As we are not experts in JAK/STAT signaling, we only looked for apparent/obvious outcomes.

3. STAT1 AD GOF phenotype (introduction p3): there are references published after ref34 to describe broader spectrum of STAT1 GOF phenotype that include impaired anti-viral defense (Toubiana et al Blood 2016). It might be helpful to cite it and modify the statement regarding anti-viral defense.

4. STAT1 GAF complex acting on a GAS promoter sequence (introduction p3): Reports with ChIP-seq/ATAC seq point to STAT binding to intronic and intergenic enhancers and consequence of STAT1 LOF on enhancers (i.e., Lau et al Nat Immunol 2018 – Fig.3). Therefore, a more updated view on ISRE and GAS motifs would include enhancers (not just promoters).

RESPONSE: We thank the reviewer for pointing us to the more recent work on JAK/STAT signaling to improve the manuscript. Both papers were included in the Introduction, and the main text was adopted accordingly.

Reviewer #2 (Remarks to the Author):

The manuscript titled "An Improved SNAP-ADAR Tool Enables Efficient RNA Base Editing to Interfere with Post-translational Protein Modification" by Kiran Kumar and colleagues is a groundbreaking study that employs RNA editing by ADAR tool for manipulating post-transcriptional modifications. The authors present an improved methodology (mainly oligo design) for this established platform that yields impressive improvements in editing efficacy as well as for the understanding of the fundamental aspects of the editing process by pointing out the validity of the "hit and run" model.

However, the main focus of the work and its importance lie in the utilization of this exciting technology for editing phosphorylation sites and demonstrating its ability to do so in dozens of sites (over 70). This is an incredibly remarkable story, as the application of RNA editing allows for more relevant PTM in comparison to current approaches, mainly due to its ability to regulate in a dose-dependent and reversible manner, which is a clear advantage over DNA editing approaches.

I find this work comprehensive, rigorous, and well-written. I believe this approach will be utilized by a large number of groups around the world and revolutionize the way we study PTMs. It is clear that works on other PTMs are on their way as well.

In summary, this paper is a groundbreaking contribution to the fields of RNA editing and PTM. One minor suggestion is to add a paragraph about the possible therapeutic benefits of this approach of PTM editing, as it is clearly the next step.

RESPONSE: We thank the reviewer for this very positive feedback on our manuscript. We added some thought on the therapeutic benefit in the last paragraph of the Discussion section.

Reviewer #3 (Remarks to the Author):

Here the authors describe improvements to their previously reported SNAP-ADAR system for directed RNA editing. This tool uses a SNAP-ADAR fusion protein and a benzylguanine-linked antisense oligonucleotide guide to direct the ADAR to the target site and form the requisite duplex for editing. In this manuscript, the authors report optimization of the antisense oligonucleotide (e.g. length, incorporation of modifications like LNA, inosine, 2'-F nucleosides and use of a bis-benzylguanine to recruit an ADAR dimer, etc) and they test different methods for delivering the SNAP-ADAR fusion protein. With the improved guide design in hand, they then apply their directed editing tool to modify codons at post translation modification sites in various signaling transcripts in HeLa cells expressing a hyperactive mutant of the SNAP-ADAR demonstrating varying editing efficiencies depending on target with some very high levels (>90%) and some very low levels (<10%). Finally, they inhibit specific PTM sites in components of the JAK-STAT signaling pathway and evaluate the impact on gene expression in response to interferon stimulation. Directed RNA editing is becoming an important method for manipulating transcriptomes with significant implications for new therapeutics. This work represents an important advance as it illustrates how improvements can be made in RNA editing tools like the SNAP-ADARs and describes a novel application of the technology (control of post translational modifications). I do have the following comments, however.

1) The authors make reference to the effect of RNA editing being fast, at one point even saying the effect is "instant", yet they show it takes 4-7 days to determine an editing phenotype at the protein level. This is not particularly fast, and it makes one wonder if this approach is best suited for probing signaling pathways mediated by much faster phosphorylation and dephosphorylations. A brief comparison to DNA editing is made at the end of the paper. However, the benefit of this approach over using, for instance, the DNA base editors developed by David Liu in a cell line like HeLa, is not clear. Would the timing of the experiment really be much different? More discussion on this comparison is justified. A potential key advantage of the use of this approach over DNA editing is the reversibility of RNA editing, but reversibility was not demonstrated in this paper.

RESPONSE: We thank the reviewer for this comment. Indeed, the text may not have been clear enough here. In cells, RNA base editing with the SNAP-ADAR tool is initiated by guide RNA transfection and takes around 6 hours to achieve the maximum editing level which stays high for several days. It then depends on the turn-over rate of the targeted protein that determines how quickly the mutation becomes fully effective. As we did not know the turn-over rate for the targeted components of the JAK/STAT pathway, we did double transfections and studied the activation of the pathway by interferon treatment a few days after the first guide RNA transfection. RNA base editing is reversible. By means of photo cleavage, one can stop the RNA base editing reaction at the RNA level instantly, however, it then takes the turn-over of the edited RNA (typically a few hours) and of the edited protein (typically few hours to a few days) to arrive fully back at the wildtype level. This was studied by us with photoactivatable and photocleavable variants of the SNAP-ADAR tool. The respective manuscript (Hanswillemenke et al., *Light-controlled RNA-Targeting with Two Self-Labeling Enzymes*) is currently under revision at a chemistry journal and is provided here for the reviewing process only. Indeed, the RNA editing process is slower than the phosphorylation switch itself and slower than the manipulation by a potential small molecule inhibitor or activator. However, compared to DNA (base) editing, RNA base editing is much faster and more straightforward. In DNA (base) editing approaches, one typically has to carry the modified cells over several passages to select and expand the genetically changed cells. RNA base editing does not require selection or passaging of the cells but is effective within hours. If a genetic change is detrimental to the cell, one

would either never collect a clone upon DNA (base) editing, or the cell's metabolism might have compensated for the genetic change and would hide a phenotype which might be visible by RNA base editing. The situation is very similar to the phenotypic differences seen when comparing gene knockdown with gene knockout. Also there, RNA interference is faster to achieve in cell culture than gene knockout and avoids genetic compensation or lethality better than the gene knockout. Both, RNA and DNA (base) editing have their respective strengths and limitations. We have rewritten the manuscript to make this clearer now. In accordance with the suggestion of the reviewer, we exchanged the term "instant" and "sudden" in the Abstract and Discussion by "fast(er)" which describes the situation (RNA versus DNA base editing) much better. Furthermore, we added more explanation in the Discussion section as suggested.

2) Are the gene expression effects only due to editing of the codon or could RNP assembly on the transcript be responsible for (at least some) the effects observed?

Along these lines, it seems imperative for an approach like this to establish that the SNAP-ADAR guided to the transcript by a high affinity oligonucleotide is not causing changes in protein expression. The authors provide a single Western blot analysis for the effect on STAT1 expression in the presence of their Y701C and T288A guides. However, there was no attempt to quantify the effects or to show reproducibility with replicate experiments.

RESPONSE: Regarding the PTMi experiment on STAT1, the reviewer might have missed that the original Figure S4 showed already a replicate. Specifically, we replicated the analysis with single guide RNA transfection, obtaining the same results as with double guide RNA transfection, indicating that the modulation of STAT1 function is fully achieved 24h after first guide transfection (see discussion about timing of the editing reaction above). So, the experiment is replicable. In the revised Supplementary Figure S4, we now provide bar graphs with the band intensities of total STAT1 and pSTAT1 under all conditions.

The reviewer is concerned that the binding of the guide RNA together with the recruited SNAP-ADAR protein could interfere with target protein expression. To address this, we designed an editing-incompetent guide RNA that carries two benzylguanine moieties to recruit two SNAP-ADAR proteins and studied its effect on the translation of the target protein. To make the guide RNA editing-incompetent, the guide was chemically modified in the central base triplet and mismatched the target adenosine with guanosine instead of cytidine. As shown in revised Supplementary Figure S4, there were no statistically significant differences in pSTAT1 protein levels of the bisBG-modified, editing-incompetent guide RNA as compared to the control lacking a guide RNA as detected via Westernblot. The same was true for the amino-guideRNA, which binds the STAT1 mRNA but does not recruit SNAP-ADAR. Experiments with the respective amino-guide RNA controls were also performed in triplicate for all four other PTMi targets of the JAK/STAT pathway and did never show inhibition of target protein translation. This also fits to the hit-and-run model (Supplementary Fig. S3) that showed that even a 20fold excess amino guide RNA was unable to interfere with the editing reaction, demonstrating that SNAP-ADAR guide RNAs spend only a short time on the target transcript.

Furthermore, the authors do not show how the other guides analyzed in Figure 5e (STAT2 Y690C, JAK1 Y1034C or IFNGR1 Y457C) effect the expression levels of those proteins. For the gene

expression changes to be ascribed only to changes in PTM of a particular protein, it must be established that the SNAP-ADAR + guide targeting a specific transcript does not alter expression levels for that protein.

This should be done in a quantitative manner with replicates and statistical analysis.

RESPONSE: In response to the reviewer's concern, we have now added a Western Blot analysis of target protein expression for all four PTMi target transcripts (STAT2, Jak1, IFNGR1, and IFNAR1) with the respective amino-guide RNA control in triplicates. In the new Supplementary Figure S5, one can clearly see that the targeting amino-guide RNA had no negative effect on target protein expression in any of the four cases compared to an empty transfection control. We applied the 2-winged t-test with unequal variance to assess potentially significant differences but did not find any. We also added a brief discussion in the main text.

REVIEWERS' COMMENTS

Reviewer #1 (Remarks to the Author):

I am satisfied with the responses provided by the authors and recommend the revised manuscript for publication in Nature Communications.

Reviewer #3 (Remarks to the Author):

The authors have addressed the concerns I raised during the original review.